# Polymerase $\theta$ is a robust terminal transferase that oscillates between three different mechanisms during end-joining

Tatiana Kent[1,2], Pedro A Mateos-Gomez[3,4], Agnel Sfeir[3,4], Richard T Pomerantz[1,2]*

[1]Fels Institute for Cancer Research, Temple University Lewis Katz School of Medicine, Philadelphia, United States; [2]Department of Medical Genetics and Molecular Biochemistry, Temple University Lewis Katz School of Medicine, Philadelphia, United States; [3]Skirball Institute of Biomolecular Medicine, New York University School of Medicine, New York, United States; [4]Department of Cell Biology, New York University School of Medicine, New York, United States

**Abstract** DNA polymerase θ (Polθ) promotes insertion mutations during alternative end-joining (alt-EJ) by an unknown mechanism. Here, we discover that mammalian Polθ transfers nucleotides to the 3′ terminus of DNA during alt-EJ in vitro and in vivo by oscillating between three different modes of terminal transferase activity: non-templated extension, templated extension *in cis*, and templated extension *in trans*. This switching mechanism requires manganese as a co-factor for Polθ template-independent activity and allows for random combinations of templated and non-templated nucleotide insertions. We further find that Polθ terminal transferase activity is most efficient on DNA containing 3′ overhangs, is facilitated by an insertion loop and conserved residues that hold the 3′ primer terminus, and is surprisingly more proficient than terminal deoxynucleotidyl transferase. In summary, this report identifies an unprecedented switching mechanism used by Polθ to generate genetic diversity during alt-EJ and characterizes Polθ as among the most proficient terminal transferases known.

*For correspondence: richard. pomerantz@temple.edu

## Introduction

DNA polymerases (Pols) are essential for life since they are necessary for the propagation and maintenance of genetic information. Intriguingly, bacterial and eukaryotic cells encode for multiple different types of Pols, some of which are intrinsically error-prone due to their relatively open active sites which enables them to tolerate particular DNA lesions (*Foti and Walker, 2010*; *Lange et al., 2011*; *Sale et al., 2012*; *Waters et al., 2009*). Such enzymes are referred to as translesion polymerases and are mostly among the Y-family of polymerases (*Foti and Walker, 2010*; *Lange et al., 2011*; *Sale et al., 2012*; *Waters et al., 2009*). Although these specialized polymerases are necessary for DNA damage tolerance, they are generally error-prone, and therefore must be highly regulated to prevent unnecessary mutations that can lead to genome instability and tumorigenesis (*Lange et al., 2011*; *Sale et al., 2012*; *Waters et al., 2009*).

The unique A-family DNA polymerase θ (Polθ), encoded by the C-terminal portion of *POLQ*, tolerates bulky lesions like Y-family polymerases and is therefore also referred to as a translesion polymerase (*Hogg et al., 2011*; *Seki et al., 2004*). However, in contrast to Y-family polymerases, Polθ is capable of replicating past the most lethal type of lesion, the double-strand break (DSB) (*Chan et al., 2010*; *Kent et al., 2015*; *Koole et al., 2014*; *Mateos-Gomez et al., 2015*; *Yousefzadeh et al., 2014*). For example, in recent studies we demonstrated the ability of the polymerase domain of *POLQ*, herein referred to as Polθ, to perform microhomology-mediated end-

**eLife digest** DNA polymerases are enzymes that replicate DNA by using single-stranded DNA as a template. DNA replication is needed to duplicate an organism's genome, and repair it if it is damaged. For example, when DNA double-strand breaks occur in the genome, DNA polymerases help repair these potentially lethal DNA breaks. If not repaired accurately, double-strand breaks in DNA can lead to genetic mutations and cancer.

Cells have evolved many different pathways to repair damaged DNA. DNA polymerase θ (called Polθ for short) is a key player in a repair mechanism called 'alternative end-joining' in mammals. In this pathway, certain enzymes trim back DNA strands on both sides of the double-stranded break to expose overhanging single strands of DNA. Polθ binds to the ends of both overhanging strands and helps them pair up with each other. Polθ then extends each single strand using the opposing overhanging strand as a template. After the gaps are filled, the DNA junction is sealed to form double-stranded DNA by other enzymes.

Previous research suggested that during alternative end-joining Polθ extends single-stranded DNA by using a guiding template strand or not using a template strand. As a consequence, Polθ frequently inserts extra pieces of DNA at the repair junction thereby introducing mutations in the DNA. It is poorly understood how this unusual DNA synthesis mechanism takes place.

Kent et al. have now investigated how Polθ extends single-stranded DNA and introduces extra DNA segments during alternative end-joining. Biochemical experiments showed that Polθ spontaneously switches between three distinct modes in which single-stranded DNA is extended. The first mode does not use a template; the second uses the opposing overhanging strand as a template; and the third uses the same overhanging strand which folds back (or snaps-back) on itself to act a template. Kent et al. also found that extra DNA pieces are inserted in all these three different modes of activity, and that this process occurs in mouse cells too. Additionally, single-stranded extension without a template was shown to be stimulated by manganese ions.

Thus by spontaneously switching between three different modes of single-stranded DNA extension, Polθ is able to incorporate diverse DNA sequence segments at DNA repair junctions. Further work is now needed to understand whether abnormal activity of Polθ contributes to cancer.

joining (MMEJ)—also referred to as alternative end-joining (alt-EJ)—in the absence of any co-factors (*Kent et al., 2015*). MMEJ requires the ability of the polymerase to perform DNA synthesis across a synapse formed between two opposing single-strand DNA (ssDNA) overhangs containing sequence microhomology (*Figure 1A*) (*Kent et al., 2015*). ssDNA overhangs are formed by partial resection of DSBs via Mre11-Rad50-Nbs1 (MRN complex) and CtIP, and potentially other factors (*Lee-Theilen et al., 2011*; *Truong et al., 2013*; *Zhang and Jasin, 2011*). Specifically, Polθ was shown to generate MMEJ products by promoting DNA synapse formation of 3' ssDNA over-hangs containing a minimal amount ($\geq$2 base pairs (bp)) of sequence microhomology, then using the opposing ssDNA overhang as a template *in trans* to extend the DNA, resulting in stabilization of the end-joining intermediate (*Figure 1A*) (*Kent et al., 2015*). The polymerase then likely extends the second overhang resulting in gap filling (*Figure 1A*). Ligase III (Lig3) is required to seal the DNA junction formed during alt-EJ/MMEJ (*Audebert et al., 2004*; *Simsek et al., 2011*), presumably after other enzymes such as endonucleases further process the end-joining intermediate (*Figure 1A*). This end-joining activity appears to be dependent on a unique insertion motif, called insertion loop 2, that also enables Polθ to bypass of other types of DNA lesions (*Hogg et al., 2011*; *Kent et al., 2015*). Thus, although Polθ is an A-family polymerase, which normally exhibit high-fidelity DNA synthesis and lack translesion synthesis activity, its unique sequence composition confers end-joining, translesion synthesis, and low-fidelity DNA synthesis activities (*Arana et al., 2008*; *Hogg et al., 2011*; *Kent et al., 2015*; *Seki et al., 2003*; *2004*).

Cellular studies show that Polθ is essential for MMEJ/alt-EJ (*Chan et al., 2010*; *Kent et al., 2015*; *Koole et al., 2014*; *Mateos-Gomez et al., 2015*; *Yousefzadeh et al., 2014*), which is consistent with biochemical studies (*Kent et al., 2015*). Intriguingly, these cellular studies showed the presence of both templated and non-templated (random) nucleotide insertions at alt-EJ repair junctions which

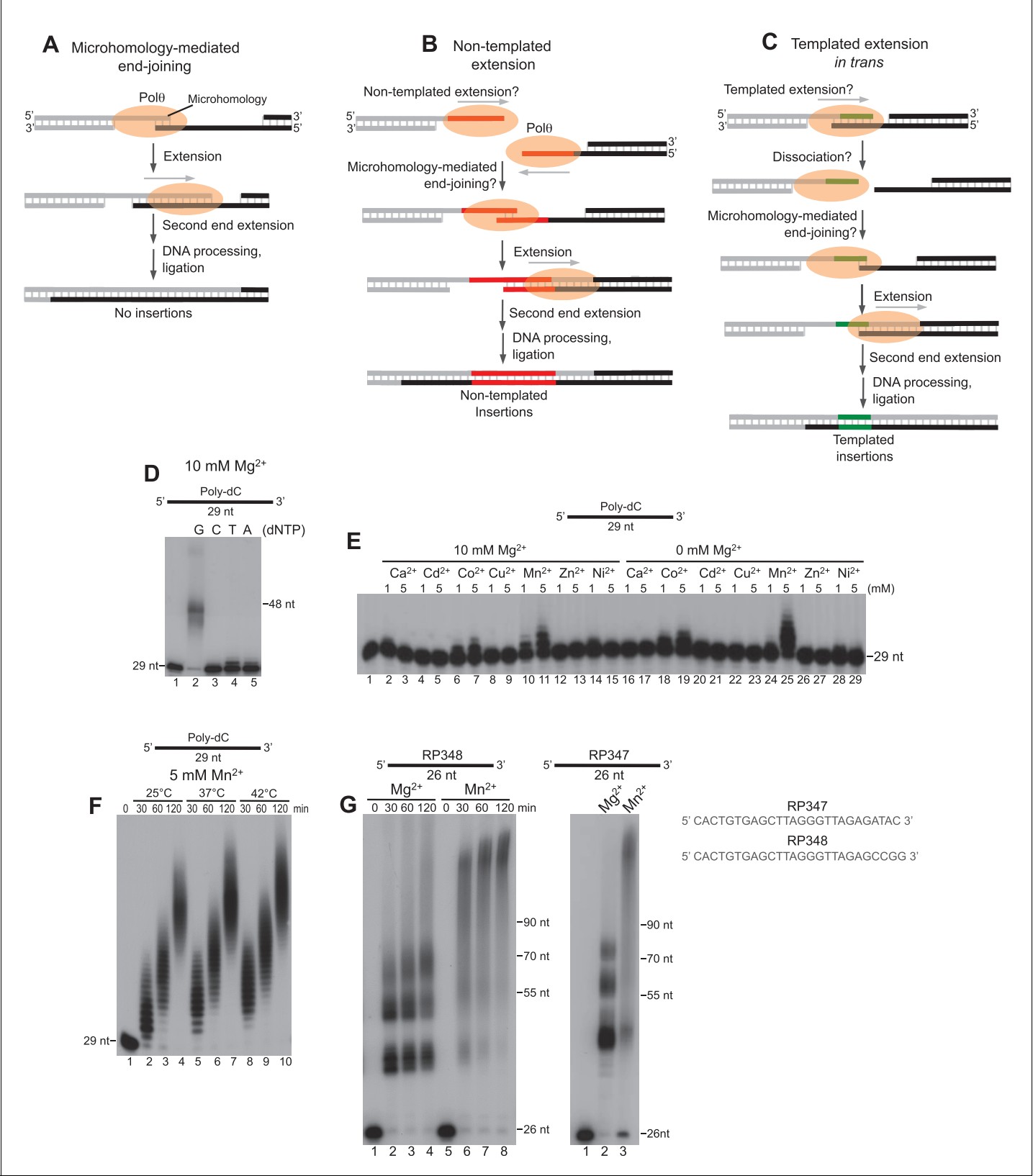

**Figure 1.** Polθ exhibits robust template-independent terminal transferase activity in the presence of manganese. (**A-C**) Models of Polθ dependent DNA end-joining. (**A**) Polθ uses existing sequence microhomology to facilitate DNA end-joining. (**B**) Polθ is proposed to extend ssDNA by a template-independent mechanism, then use the newly generated sequence to facilitate DNA end-joining. (**C**) Polθ is proposed to extend ssDNA by using the

*Figure 1 continued on next page*

*Figure 1 continued*

opposing overhang as a template *in trans*, then after DNA synapse dissociation Polθ uses the newly generated sequence to facilitate DNA end-joining. (D) A denaturing gel showing Polθ extension of poly-dC ssDNA in the presence of indicated dNTPs and 10 mM $Mg^{2+}$. (E, F) Denaturing gels showing Polθ extension of poly-dC ssDNA in the presence of dTTP and indicated divalent cation concentrations (E) and time intervals and temperatures (F). (G) Denaturing gels showing Polθ extension of indicated ssDNA in the presence of all four dNTPs and 10 mM $Mg^{2+}$ or 5 mM $Mn^{2+}$.

The following figure supplements are available for figure 1:

**Figure supplement 1.** Polθ template-independent activity is stimulated by physiological concentrations of $Mn^{2+}$ and $Mg^{2+}$.

**Figure supplement 2.** Optimization of Polθ-Mn2+ template-independent terminal transferase activity.

**Figure supplement 3.** Sequence analysis of Polθ-$Mg^{2+}$ template-dependent terminal transferase activity.

were dependent on Polθ (*Chan et al., 2010*; *Koole et al., 2014*; *Mateos-Gomez et al., 2015*; *Yousefzadeh et al., 2014*). The random insertions were suggested to be due to a putative Polθ template-independent activity (*Figure 1B*) (*Mateos-Gomez et al., 2015*). Yet, insofar Polθ template-independent terminal transferase activity has not been demonstrated in vitro. For example, early in vitro studies showed the unusual ability of Polθ to extend ssDNA and partial ssDNA substrates with 3' overhangs (pssDNA) by several nucleotides (*Hogg et al., 2012*). Although it was suggested that this activity might be the result of template-independent terminal transferase activity, the polymerase failed to extend homopolymeric ssDNA templates, which contain one type of base, without the complementary deoxyribonucleoside-triphosphate (dNTP) present (*Hogg et al., 2012*). These previous studies therefore demonstrated a lack of template-independent terminal transferase activity by Polθ (*Hogg et al., 2012*). More recent studies confirmed a lack of template-independent activity by Polθ (*Yousefzadeh et al., 2014*). Instead, data was presented that suggests Polθ extends ssDNA by transiently annealing two oligonucleotides together in an anti-parallel manner, resulting in repeated use of the opposing ssDNA as a template *in trans* (*Yousefzadeh et al., 2014*). Since cellular studies additionally demonstrated the presence of templated nucleotide insertions at alt-EJ junctions, which were also dependent on expression of the polymerase, models of Polθ copying sequences from the opposing overhang were also proposed (*Figure 1C*) (*Chan et al., 2010*; *Koole et al., 2014*; *Yousefzadeh et al., 2014*). This form of templated extension *in trans* can conceivably facilitate end-joining by generating short regions of microhomology (*Figure 1C*). Although previous biochemical studies suggested the possibility that Polθ performs templated extension of ssDNA *in trans* (*Yousefzadeh et al., 2014*), more recent studies showed that the polymerase extends ssDNA by performing 'snap-back' replication on the same template *in cis*, similar to other end-joining polymerases (*Brissett et al., 2007*; *Kent et al., 2015*).

Clearly, our understanding of how Polθ extends ssDNA, which is important for alt-EJ and is a unique activity for this polymerase, is very limited. For example, it remains to be determined whether Polθ is capable of performing non-templated extension (*Figure 1B*), and templated extension *in trans* in the absence of sufficient microhomology (*Figure 1C*). Furthermore, whether other factors or co-factors are necessary for activating Polθ terminal transferase activity during alt-EJ, which likely facilitates insertion mutations, remains unknown. Considering that Polθ contributes to the survival of breast and ovarian cancer cells deficient in homologous recombination (HR), is associated with a poor clinical outcome for breast cancer patients, and confers resistance to chemotherapy drugs and ionizing radiation, understanding the enzymatic functions of the polymerase is essential for elucidating its roles in cancer progression and chemotherapy resistance (*Ceccaldi et al., 2015*; *Higgins et al., 2010*; *Lemee et al., 2010*; *Mateos-Gomez et al., 2015*).

In this study, we sought to elucidate how Polθ generates insertion mutations during alt-EJ which contribute to genome instability. First, we found that manganese ($Mn^{2+}$) activates Polθ template-independent terminal transferase activity. Next, we discovered that Polθ generates random combinations of templated and non-templated insertion mutations during alt-EJ by oscillating between three different modes of terminal transferase activity: non-templated extension, templated extension *in cis*, and templated extension *in trans*. Lastly, we further characterized Polθ terminal transferase activity and surprisingly found that this activity is more proficient than terminal deoxynucleotidyl

transferase (TdT). Together, these data identify an unprecedented switching mechanism employed by Polθ to generate genetic diversity during alt-EJ and characterize Polθ as among the most proficient terminal transferases in nature.

## Results

### Polθ template-independent activity requires manganese

A current paradox in our understanding of alt-EJ is that Polθ promotes non-templated (random) nucleotide insertions at DNA repair junctions in vivo, but lacks template-independent terminal transferase activity in vitro. For example, similar to previous studies (*Hogg et al., 2012*; *Kent et al., 2015*), we found that Polθ fails to extend a homopolymeric ssDNA containing deoxycytidine-monophosphates (poly-dC) in the absence of the complementary deoxyguanosine-triphosphate (dGTP) under standard buffer conditions with magnesium ($Mg^{2+}$ *Figure 1D*). This shows that efficient ssDNA extension by Polθ requires the complementary nucleotide, which demonstrates that the template bases facilitate the nucleotidyl transferase reactions by pairing with the incoming nucleotide. Our recent studies suggest that this template-dependent activity is due to 'snap-back' replication whereby the polymerase uses the template *in cis* (*Kent et al., 2015*). A separate biochemical study also indicated that Polθ lacks template-independent activity (*Yousefzadeh et al., 2014*). Thus, it remains unclear how Polθ facilitates random nucleotide insertions during alt-EJ which contribute to genome instability (*Figure 1B*).

Considering that divalent cations other than $Mg^{2+}$ are present in cells, they may account for the discrepancy between the ability of Polθ to perform template-independent DNA synthesis in vivo but not in vitro. We therefore tested various divalent cations in a reaction including Polθ, poly-dC ssDNA and deoxythymidine-triphosphate (dTTP), in the presence and absence of $Mg^{2+}$ (*Figure 1E*). The results showed that $Mn^{2+}$, and to a lesser extent $Co^{2+}$, activates Polθ extension of poly-dC with dTTP (*Figure 1E*). For example, in the absence of $Mn^{2+}$ in *Figure 1D*, Polθ extended only a small fraction of substrates with dTTP (lane 4). In contrast, the addition of $Mn^{2+}$ under the same reaction conditions promoted extension of the same substrate by Polθ even when $Mg^{2+}$ was abundant (*Figure 1E*). Since thymidine cannot base pair with cytidine, these data demonstrate that $Mn^{2+}$ activates Polθ template-independent terminal transferase activity (i.e. non-templated DNA synthesis). Since Polθ DNA synthesis activity is fully supported by $Mn^{2+}$ (*Figure 1E*, lane 25), this indicates that $Mn^{2+}$ binds to the same positions as $Mg^{2+}$ within the polymerase active site which is necessary for the nucleotidyl transferase reaction. Consistent with this, recent structural studies show that other metals such as calcium can substitute for $Mg^{2+}$ in the polymerase active site (*Zahn et al., 2015*). Furthermore, several lines of evidence show that $Mn^{2+}$ can act as a co-factor for DNA polymerases and RNA polymerases and reduces the fidelity of these enzymes (*Andrade et al., 2009*; *Dominguez et al., 2000*; *Walmacq et al., 2009*). Hence, the data show that $Mn^{2+}$ acts as a co-factor for Polθ which promotes template-independent activity and likely reduces the fidelity of the polymerase. Importantly, this template-independent activity was also stimulated 3–8 fold by relatively low concentrations of $Mn^{2+}$ (0.2 mM) and $Mg^{2+}$ (1–2 mM) which are found in cells (*Figure 1—figure supplement 1*) (*MacDermott, 1990*; *Schmitz et al., 2003*; *Visser et al., 2014*). Biochemical studies have also shown that $Mn^{2+}$ is a necessary co-factor for the yeast Mre11-Rad50-Xrs2 (MRX) nuclease complex and its mammalian counterpart, MRN, which is essential for generating 3' overhangs during alt-EJ, presumably by acting with CtIP (*Lee-Theilen et al., 2011*; *Trujillo et al., 1998*; *Zhang and Jasin, 2011*). Thus, these and other lines of evidence strongly indicate a physiological role for $Mn^{2+}$ as a co-factor for DNA repair enzymes (*Andrade et al., 2009*; *Cannavo and Cejka, 2014*; *Dominguez et al., 2000*; *Trujillo et al., 1998*).

We identified optimal conditions for Polθ-$Mn^{2+}$ template-independent terminal transferase activity in *Figure 1—figure supplement 2*. Using these optimal conditions at different temperatures, we found that Polθ-$Mn^{2+}$ exhibits robust template-independent terminal transferase activity (*Figure 1F*). This suggests $Mn^{2+}$ promotes the ability of Polθ to generate random nucleotide insertions during alt-EJ in cells. We further found that $Mn^{2+}$ greatly stimulates Polθ terminal transferase activity on non-homopolymeric ssDNA substrates (*Figure 1G*, left and right). In contrast, in the presence of $Mg^{2+}$ Polθ became mostly arrested after transferring ~10–20 nucleotides (nt), but also generated some larger discrete products (*Figure 1G*, left and right). These data along with those presented in

*Figure 1D* indicate that $Mg^{2+}$ promotes template-dependent activity which directs the polymerase to repeatedly synthesize a few discrete products as observed for both substrates (*Figure 1G*, left and right). Consistent with this, we found that Polθ-$Mg^{2+}$ consistently generated similar DNA sequences from the RP347 ssDNA template, which is likely due to snap-back replication (*Figure 1—figure supplement 3*). $Mn^{2+}$ on the other hand facilitates template-independent activity which enables Polθ to generate random products of different lengths as indicated by a smear (*Figure 1G*, left and right).

## Polθ oscillates between three different modes of terminal transferase activity

To gain more insight into these mechanisms of Polθ terminal transferase activity, we analyzed the sequences of ssDNA extension products generated by Polθ-$Mn^{2+}$ in the absence of $Mg^{2+}$ and with a 10-fold excess of $Mg^{2+}$ which models cellular conditions. As expected, most of the DNA sequence generated by Polθ-$Mn^{2+}$ in the absence of $Mg^{2+}$ was random and therefore due to template-independent activity (*Figure 2A*). This is consistent with the appearance of a smear rather than a few discrete bands as observed with Polθ-$Mg^{2+}$ (*Figure 1G*). Intriguingly, some of the sequences contained short regions that were either identical or complementary to the initial ssDNA (*Figure 2A*, black underlines). Other sequence regions within individual molecules were complementary to one another but not to the original ssDNA template (*Figure 2A*, grey and colored lines). Next, we analyzed DNA sequences generated by Polθ in the presence of a ten-fold excess of $Mg^{2+}$ relative to $Mn^{2+}$, which more closely resembles physiological conditions (*Figure 2B*). Again, we observed random sequence, complementary sequences within individual products (grey and colored lines), and short sequence tracts identical or complementary to the initial template (black underlines). Interestingly, Polθ generated more complementary sequences with an excess of $Mg^{2+}$ (compare *Figure 2A and B*). Furthermore, the average length of ssDNA extension products was shorter with an excess of $Mg^{2+}$ (*Figure 2E*), which is consistent with the results in *Figure 1G*.

Together, these data demonstrate that Polθ exhibits three distinct modes of terminal transferase activity when $Mn^{2+}$ is present even at 10-fold lower concentrations than $Mg^{2+}$ (*Figure 2C*). In the first and predominant mode, Polθ performs template-independent terminal transferase activity (*Figure 2C*, top). In the second mode, Polθ performs transient template-dependent extension *in cis*, also called snap-back replication (*Figure 2C*, bottom left). This mechanism accounts for the appearance of complementary sequences within individual extension products (*Figure 2A,B*; grey and colored lines). In the third mode, Polθ performs transient template-dependent extension *in trans* (*Figure 2C*, bottom right). This accounts for sequence tracts that are identical or complementary to the initial ssDNA substrate (*Figure 2A,B*; black underlines); templated extension *in cis* can also promote sequence complementary to the initial template (*Figure 2C*). Identical sequence tracts are most likely due to copying *in trans* of complementary sequence tracts initially formed by templated extension *in cis* or *in trans* (*Figure 2—figure supplement 1*). Further in vitro and in vivo evidence for these three mechanisms of terminal transferase activity is presented in *Figures 3* and *4*, respectively.

Intriguingly, many of the extension products were generated by more than one mode of terminal transferase activity (*Figure 2B*), which demonstrates that the polymerase oscillates between these different mechanisms (*Figure 2C*). We utilized product sequences to specifically trace this enzymatic switching phenomenon at near base resolution (*Figure 2D*). For example, sequence 8 from *Figure 2B* demonstrates that Polθ first performs 50 consecutive random nucleotide transfer events, then switches to a transient snap-back replication mode (templated extension *in cis*). Next, Polθ switches to random mode then after transferring 4 nt switches back to snap-back mode followed by another switch back to random synthesis. Next, Polθ switches to the templated extension *in trans* mode where it copies 7 nt, then switches back to random mode for an additional 23 nt. Finally, Polθ switches back to snap-back mode, then after transferring 8 nt it ends the reaction by randomly incorporating an additional 5 nt. Sequence 3 from *Figure 2B* shows similar oscillation between these different mechanisms (*Figure 2D*, bottom). Here, Polθ performs 55 consecutive random nucleotide transfer events then switches to snap-back mode where it incorporates another 15 nt. Since the melting temperature of this 15 bp duplex is predicted to be 50°C and the reaction was performed at 42°C, Polθ appears to be capable of unwinding duplexes formed during snap-back replication. Polθ then performs three additional switching events, ultimately generating in a 138 nt product composed of a combination of random and templated sequence.

**A** 5 mM Mn²⁺  5' CACTGTGAGCTTAGGGTTAGAGATAC 3'

1 AATCTAATCCTGATCACTGTGAGTGATTCGGGTAGATTCTTAGTGTGGAATGTTTACACTATCTACTTAGCGTAATAGGCTGAGTGTTTTTATATAGCAAACCTCTGAACTTAGGTTTGTT

2 TAAACATCACCTGTTTAGATCTCTCAGTAGAGGGGGTGACTAAGGAGATCCAGAAGACTTAGGTGAATCACCTAAGTCTTCTGGTTTAGTTTAGCTTAGGAAATTTAAAACATAAGCTAAAACCAATCTTGCTAGTGTTATAGTTATTTAATAGTCTTAAGTAGTGATTCTAATGTTTCAAAAACGTAAAAATGCCTCCTTGGCAGAGACGAGATTCTACGCGAAGCTCCCTCGCTGGATCTTGAGGTGAGTCGGTGCTACTAAAGACG

3 AGACCTAAGCTCATGTGAGTGATGTCACACTGACAACCACAGATTGACGTCAATGTGTCATTATACGATGACGCTTTCTATAATGACGTTTAATTCTTTAGCTGTATGTTAGTACGGCTGATGTGTACTGTCTCAGACCGTGCAACGGTCTTAGATGGGCATCGTACACATGAGAGGGGTAAGTTCTTTTGCGCTCAGGACCAT

4 GGTGGTGTGTATTTCACTCTCTTACACGCTCATAGGTGTGAAGTGAAGATGTCAGTAAGGGATTTACTATTAGCTCTATGAGCGTGTGGAGGGAGTGA

5 GTAGTGAAGGCCTGTCTTCAACAGAAAGATTTCGGTTGAGTAGGTTGGTTGGGCTATCAAATATCTACTTCGACTACCACGACCGTAGAAAGTAAGGGTAGGCCGAA

6 TCGGTGAAGTTGTATTTCTTCATATTAGTAAAATATCACTGTCTTTCTGACGTGAAGGAGATTTCTCTCCAGAGAGTGTGAGAGTAATAT

7 AGTGCAGTAT

8 CTGAAATGGTAACCATTTTCAGGGTCGCCCTGAGTGAGGACGACACTCGGGGGCTACCCT

9 AGTAGTGTGTGTTTTAACGCGTGCTCAAGACATAGAGGACTGTTGAGCTCTGTACAGCTTACTGAAGCTGCGTACGTGAGTACGCAGGCCAGTAGAGAGAGTCACCTTACGGGTAATTTAAGGGACCTGGCCACTGCATGGCCAT

**B** 10 mM Mg²⁺, 1 mM Mn²⁺  5' CACTGTGAGCTTAGGGTTAGAGATAC 3'

1 AGAGGTATTCTAGCCACGCTCACGGTGAGT

2 CCTAGGCCTCACGGTGAAATGACTTTTCACTTCAAGTGAAGTGGAAAAGTCATATTCACCCGTATGGGGT

3 CTGAAGTGATATCTCATCTCCAGGTTATATATGGGGGTGCATTCCCCGTAAATGTGAGATGAGGTATCACGTTTCTCGTGATGCATCATGTTTATGAGATGAAGCATATTGACTCTTGGAGTTCAGTTCTACGGGTAT

4 TCGGTGAAGTTGTATTTCTTCATATTAGTAAAATATCACTGTCTTTCTGACGTGAAGGAGATTTCTCTCCAGAGAGTGTGAGAGTAATAT

5 CCTTATATCTGT

6 CTAATCCTTAGTTTAGAAGGGCTATCTTCTAAAACTAAGGATT

7 ATACTGCACT

8 CCGAAAGGGGATTTCTATACCCTGAAGTCGCAGTGCGACTTCGGGGTGTGGAAATCCACCTTTCAGGGCTACCCTAGGTATGTAGGGTTGGCCTTGAAGTGAAGCCGGGCGACCTACATATTCAT

9 AATCTAATCCTGATCACTGTGAGTGATTCGGGTAGATTCTTAGTGTGGAATGTTTACACTATCTACTTAGCGTAATAGGCTGAGTGTTTTTATATAGCAAACCTCTGAACTTAGGTTTGTT

**C** Non-templated extension

Palm
Mg²⁺ Mn²⁺
Primer
5'
Fingers
Thumb

Thumb
Primer
5'
Fingers
Palm

Templated extension *in cis*
(Snap-back replication)

Palm
3'
Primer
5'
Fingers
Thumb
Template
*in trans*
5'

Templated extension *in trans*

**D** Sequence B8

5' Non-templated extension (50 nt)
5' Templated extension *in cis* (7 nt)
5' Non-templated extension (4 nt)
5' Templated extension *in cis* (7 nt)
5' Non-templated extension (14 nt)
5' Templated extension *in trans* (7 nt)
5'
5' Non-templated extension (23 nt)
5' Templated extension *in cis* (8 nt)
5' Non-templated extension (5 nt)
5'

Sequence B3

5' Non-templated extension (55 nt)
5' Templated extension *in cis* (15 nt)
Tm = 50° C
DNA unwinding
5' Non-templated extension (23 nt)
5' Templated extension *in cis* (7 nt)
5' Non-templated extension (38 nt)
5'

**E**

ssDNA extension products

Avg. length
Avg. length

Mn²⁺ + Mg²⁺
Mn²

ssDNA length (nt)
0   50   100   150   200   250   300   350

**Figure 2.** Polθ oscillates between three different modes of terminal transferase activity. (A,B) Sequences of Polθ ssDNA extension products in the presence of indicated divalent cations (**A**, 5 mM Mn$^{2+}$; **B**, 10 mM Mg$^{2+}$, 1 mM Mn$^{2+}$). Initial ssDNA sequences are indicated at top. Black underlines, sequences copied from either original template or complementary sequences generated from original template; matching colored lines, complementary sequences due to snap-back replication. (C) Models of Polθ terminal transferase activities. (Top) Polθ preferentially exhibits template-independent activity in the presence of Mg$^{2+}$ and Mn$^{2+}$. Polθ also performs templated ssDNA extension *in cis* (bottom left) and *in trans* (bottom right), and oscillates between these three mechanisms. (D) Models of Polθ terminal transferase activity based on sequences 3 and 8 from panel B. (E) Plot showing lengths of ssDNA products generated by Polθ in the presence of indicated divalent cations.

The following figure supplements are available for figure 2:

**Figure supplement 1.** Model of Polθ-Mn$^{2+}$ terminal transferase activity involving template copying *in cis* and *in trans*.

**Figure supplement 2.** Control experiments for Polθ-Mn$^{2+}$ template-independent activity.

**Figure supplement 3.** Polθ-Mn$^{2+}$ exhibits *de novo* DNA and RNA synthesis activities.

**Figure supplement 4.** Polθ-Mn$^{2+}$ exhibits processive terminal transferase activity.

**Figure supplement 5.** Polθ-Mn$^{2+}$ oscillates between different terminal transferase activites in the presence of a DNA trap.

**Figure supplement 6.** Polθ oscillates between templated and non-templated terminal transferase activities in the presence of physiological concentrations of Mg$^{2+}$ and Mn$^{2+}$.

Under these conditions, Polθ shows a preference for template-independent terminal transferase activity (*Figure 2C*), which is more prevalent when Mg$^{2+}$ is omitted (compare *Figures 2A and B*). Thus, the ratio of Mn$^{2+}$ to Mg$^{2+}$ modulates the balance between these different mechanisms. For example, higher concentrations of Mn$^{2+}$ promote template-independent transfer events, whereas lower concentrations of Mn$^{2+}$ reduce random transferase activity while increasing template-dependent activity due to snap-back replication (compare *Figures 2A and B*). Higher concentrations of Mn$^{2+}$ also promote longer extension products, which correlates with the polymerase's preference for template-independent activity under these identical conditions (*Figure 1G*; *Figure 2E*).

To be certain Polθ-Mn$^{2+}$ performs template-independent activity rather than highly error-prone template-dependent activity which may be perceived as template-independent, we performed multiple additional controls. First, we analyzed template- dependent and independent activities in the same reaction performed in solid-phase (*Figure 2—figure supplement 2A,B*). Here, a biotinylated primer-template was immobilized to streptavidin beads, then excess template strand was removed by thorough washing. Primer extension in the presence of Mn$^{2+}$ was then performed and extension products were sequenced. The results show that the initial template-dependent activity is performed with relatively high fidelity (*Figure 2—figure supplement 2B*). For example, misincorporation and frameshift error rates of $5.6 \times 10^{-2}$ and $6.9 \times 10^{-3}$, respectively, were observed on this short template. On the other hand, once Polθ reaches the end of the template mostly random sequence was generated, demonstrating template-independent activity (*Figure 2—figure supplement 2B*). Consistent with this, we show the ability of Polθ to continue DNA synthesis far beyond the end of the template exclusively in the presence of Mn$^{2+}$ (*Figure 2—figure supplement 2C*). We further show that the rate of misincorporation and mismatch extension by Polθ-Mn$^{2+}$ on a primer-template in the presence of a single nucleotide (dATP) is dramatically slower than its activity under identical conditions without the template strand present (*Figure 2—figure supplement 2D*). Thus, these data demonstrate that Polθ-Mn$^{2+}$ terminal transferase activity is not the result of misincorporation or mismatch extension. As an additional control for template-independent activity, we tested whether Polθ-Mn$^{2+}$ performs *de novo* synthesis in the absence of DNA. Remarkably, Polθ-Mn$^{2+}$ exhibits *de novo* DNA and RNA synthesis which unequivocally demonstrates its ability to synthesize nucleic-acids in a template-independent manner (*Figure 2—figure supplement 3*).

Next, we examined whether Polθ-Mn$^{2+}$ acts processively during ssDNA extension and whether the polymerase can switch between the three different modes of terminal transferase activity without dissociating from the initial ssDNA template. We tested the processivity of Polθ-Mn$^{2+}$ on ssDNA by allowing the polymerase to extend the ssDNA for an initial 5 min followed by the addition of a 150-

fold excess of unlabeled ssDNA which sequesters the polymerase if it dissociates from the initial radio-labeled ssDNA during the reaction (*Figure 2—figure supplement 4B*). Remarkably, addition of the ssDNA trap had no effect on Polθ-Mn$^{2+}$ terminal transferase activity, demonstrating that the polymerase performs ssDNA extension with high processivity. As a control, we show that 150-fold excess of unlabeled ssDNA effectively sequesters the polymerase from solution (*Figure 2—figure supplement 4A*). Since Polθ-Mn$^{2+}$ exhibits three different modes of terminal transferase activity under the same conditions (*Figure 2A*), these results indicate the polymerase switches between these distinct activities without dissociating from the initial ssDNA.

To further test the processivity of this switching mechanism, we performed ssDNA extension in the presence and absence of a ssDNA trap in solid-phase which enabled removal of excess unbound polymerase from solution (*Figure 2—figure supplement 5*). For example, Polθ was first allowed to bind ssDNA immobilized to streptavidin beads. Then, excess unbound Polθ was removed by thorough washing of the beads. Next, the reaction was initiated by the addition of dNTPs in buffer containing 10 mM Mn$^{2+}$ and 1 mM Mn$^{2+}$. After 15 s, a 150-fold excess of ssDNA trap was added, whereas the negative control reaction contained no trap. Following completion of the reactions, the immobilized ssDNA was isolated and sequenced. Consistent with the results obtained in *Figure 2—figure supplement 4*, the ssDNA trap did not suppress Polθ terminal transferase activity. In fact, the data indicate that the addition of excess ssDNA increases the length of ssDNA extension products generated by Polθ in solid-phase (*Figure 2—figure supplement 5*, panels B–D). This suggests that use of a template *in trans* enables Polθ terminal transferase activity rather than suppressing it. Consistent with this, sequence analysis shows that Polθ frequently utilizes the ssDNA trap as a template *in trans* (*Figure 2—figure supplement 5*, red underlines, panel D). The polymerase also performs template-independent and snap-back replication activities when the ssDNA trap is present (panel D). Since Polθ is highly processive during ssDNA extension (*Figure 2—figure supplement 4*), these data provide strong support for a model whereby a single polymerase oscillates between the three different modes of terminal transferase activity without dissociating from the initial ssDNA template. Importantly, using intracellular concentrations of Mg$^{2+}$ (1 mM) and Mn$^{2+}$ (50 µM), Polθ remains effective in extending ssDNA and utilizes a combination of templated and non-templated mechanisms during this activity (*Figure 2—figure supplement 6*).

## Polθ oscillates between three modes of terminal transferase activity during alt-EJ

Next, we examined Polθ terminal transferase activity in the context of alt-EJ. Although cellular studies have shown that Polθ expression is required for the appearance of non-templated and templated insertions at alt-EJ repair junctions, it remains unknown whether additional factors or co-factors facilitate these insertion events. For example, Polθ has been shown to promote what appears to be random nucleotide insertion tracts at alt-EJ repair junctions in mice and flies (*Figure 1B*) (*Chan et al., 2010*; *Mateos-Gomez et al., 2015*). Evidence in flies, mice and worms also indicates that Polθ promotes templated nucleotide insertions, which are proposed to be due to a template copy mechanism *in trans* (*Figure 1C*) (*Chan et al., 2010*; *Koole et al., 2014*). To determine whether Polθ is solely responsible for these insertions, and whether the three mechanisms of terminal transferase activity identified herein facilitate these insertions, we reconstituted a minimal alt-EJ system in vitro. Here, two DNA substrates containing a 3' overhang, herein referred to as partial ssDNA (pssDNA), and a single base pair of microhomology (G:C) at their 3' termini were incubated with Polθ, Lig3, ATP, and dNTPs in buffer containing a high ratio of Mg$^{2+}$ to Mn$^{2+}$ which models cellular conditions (*Figure 3A*, top). Although Polθ can perform MMEJ without Lig3 by promoting templated extension *in trans* (*Figure 1A*) (*Kent et al., 2015*), the pssDNA substrates in the current assay lack sufficient microhomology for MMEJ, but contain a 5' phosphate on their short strands which can support ligation of the opposing 3' overhang that is extended by the polymerase (*Figure 3A*, top). Control experiments show that the addition of Polθ and Lig3 is required for efficient alt-EJ, and that insertions depend on Polθ (*Figure 3—figure supplement 1C*). These results are expected since Lig3 is required for most alt-EJ in cells and therefore likely functions with Polθ which facilitates insertions (*Audebert et al., 2004*; *Simsek et al., 2011*). Following termination of the reaction by EDTA, DNA was purified then end-joining products were amplified by PCR and individually sequenced from cloning vectors (*Figure 3—figure supplement 1A,B*).

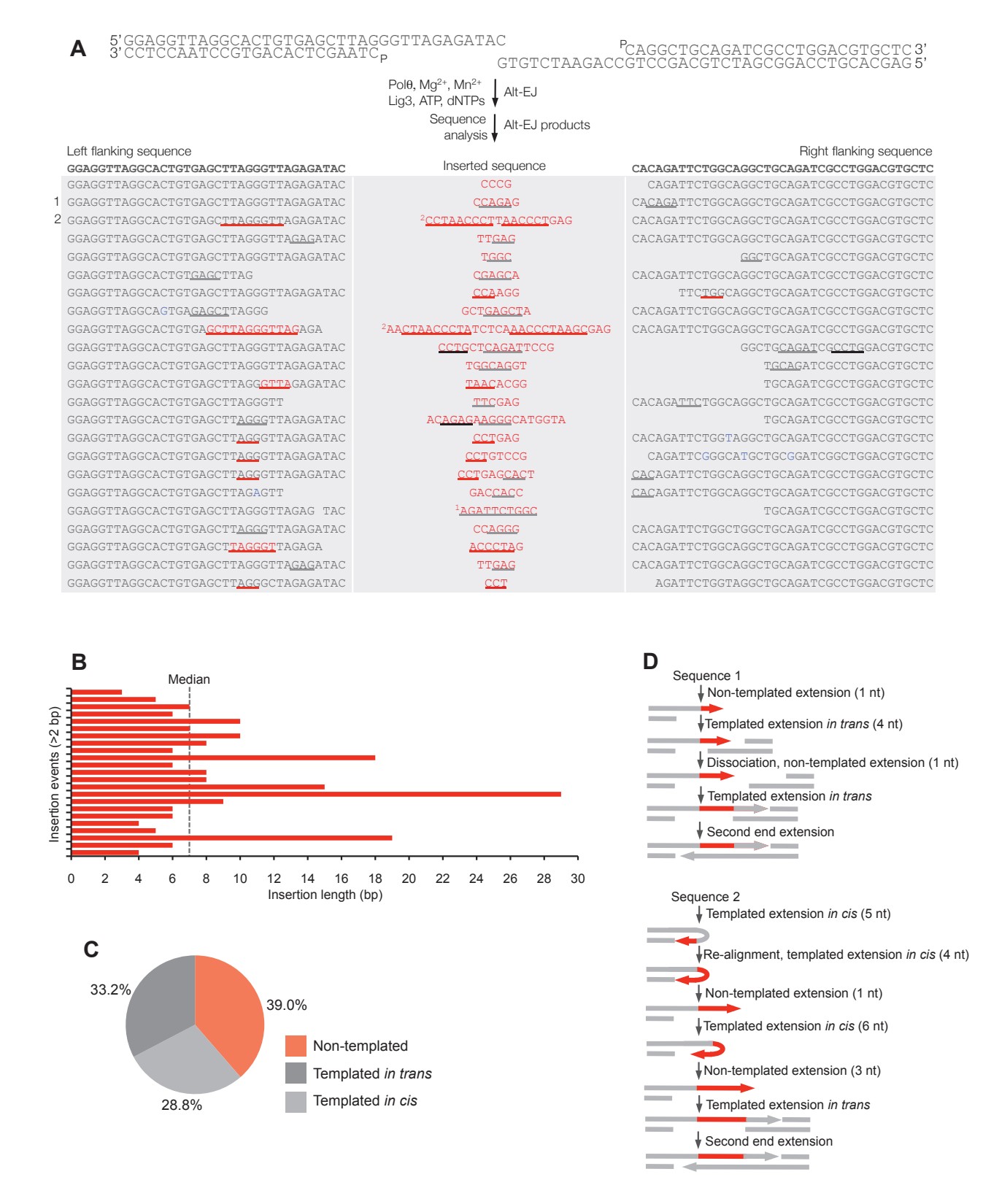

**Figure 3.** Polθ oscillates between three different modes of terminal transferase activity during alternative end-joining in vitro. (**A**) Scheme for reconstitution of Polθ mediated alt-EJ in vitro (top). Sequences of alt-EJ products generated by Polθ in vitro using 10 mM Mg²⁺ and 1 mM Mn²⁺

*Figure 3 continued on next page*

*Figure 3 continued*

(bottom). Red text, insertions; black text, original DNA sequence; black and grey underlines, sequences copied from original template; red underlines, complementary sequences due to snap-back replication; red sequence without underlines, random insertions; superscript 1, suggests sequences were copied from a template portion that was subsequently deleted during alt-EJ; superscript 2, suggests sequences were copied from the template in more than one way. Original DNA sequences indicated at top. Blue type, mutations. (**B**) Plot of insertion tract lengths generated in panel **A**. (**C**) Chart depicting percent of individual nucleotide insertion events due to non-templated extension, templated extension *in cis* and templated extension *in trans. t* test indicates no significant difference between percent of non-templated and templated *in cis* insertions. (**D**) Models of Polθ activity based on end-joining products 1 and 2 from panel **A**.

The following figure supplements are available for figure 3:

**Figure supplement 1.** Supporting information for Polθ mediated alt-EJ in vitro.
**Figure supplement 2.** Polθ acts processively during alt-EJ in vitro.
**Figure supplement 3.** Polθ generates insertions during alt-EJ in the presence of low concentrations of Mg2+ and Mn2+.

To gain significant insight into the mechanisms of Polθ terminal transferase activity during alt-EJ, we chose to analyze insertion tracts greater than 2 nt in length which reveal information regarding template dependency. Remarkably, we found that Polθ generated both random and templated nucleotide insertions at repair junctions (*Figure 3A*), which is similar to the results obtained in *Figure 2*. In the case of templated insertions, we observed sequence tracts that appear to be due to both templated extension *in cis* (snap-back replication; red underlines) and *in trans* (grey underlines). A median insertion length of 7 bp was observed (*Figure 3B*), and cumulative analysis of individual nucleotide insertion events reveals a roughly equal proportion of insertions due to the three modes of terminal transferase activity identified in *Figure 2*, for example non-templated extension, templated extension *in cis*, and templated extension *in trans* (*Figure 3C*). We again modeled Polθ switching activity based on the sequence generated, in this case during alt-EJ (*Figure 3D*). Consistent with the mechanism identified in *Figure 2*, sequence traces strongly suggest spontaneous and rapid switching between the three different terminal transferase activities (*Figure 3D*).

We next examined whether the polymerase acts processively to generate insertions during alt-EJ. To test this, we repeated the alt-EJ reaction in vitro, but added a 150-fold excess of ssDNA trap 15 s after the reaction was initiated. The results show that Polθ generates similar insertion tract lengths in the presence and absence of the ssDNA trap (compare *Figure 3* and *Figure 3—figure supplement 2*). Thus, these data also indicate that Polθ acts processively during alt-EJ which provides further support for a model whereby a single polymerase oscillates between the different terminal transferase activities prior to dissociating from the initial substrate. Importantly, further alt-EJ experiments show that Polθ generates similar size insertions by a combination of templated and non-templated mechanisms in the presence of 1 mM $Mg^{2+}$ and 50 µM $Mn^{2+}$ which model intracellular concentrations (*Figure 3—figure supplement 3*).

To test whether Polθ uses this switching mechanism to generate insertions during alt-EJ in cells, we analyzed insertion tracts synthesized by Polθ during alt-EJ in vivo (*Figure 4*). Here, Polθ dependent alt-EJ in mouse embryonic stem cells promotes translocations between sequence specific DSBs generated in chromosomal DNA by the CRISPR/Cas9 system, as shown in previous studies (*Figure 4A*, top *Mateos-Gomez et al., 2015*). To distinguish between the different Polθ mediated activities during chromosomal translocation, we carefully analyzed junctions of events resulting from the cleavage of chromosomes 6 and 11, and subsequent formation of Der (6) and (11). Similar to *Figure 3*, we chose to analyze junctions containing insertions >2 bp in length. Remarkably, in the cellular alt-EJ system we also observed insertion tracts that appear to be due to all three modes of Polθ terminal transferase activity (*Figure 4A*). For example, similar to the results obtained in the in vitro alt-EJ system (*Figure 3*), cumulative analysis of individual nucleotide insertion events produced in vivo demonstrates that Polθ generates a roughly equal proportion of insertion events due to the three different modes of terminal transferase activity (*Figure 4A–C*). Templated extension *in trans* accounts for short sequence duplications (black and grey underlines), whereas templated extension *in cis* (snap-back replication) accounts for the appearance of short complementary sequence tracts

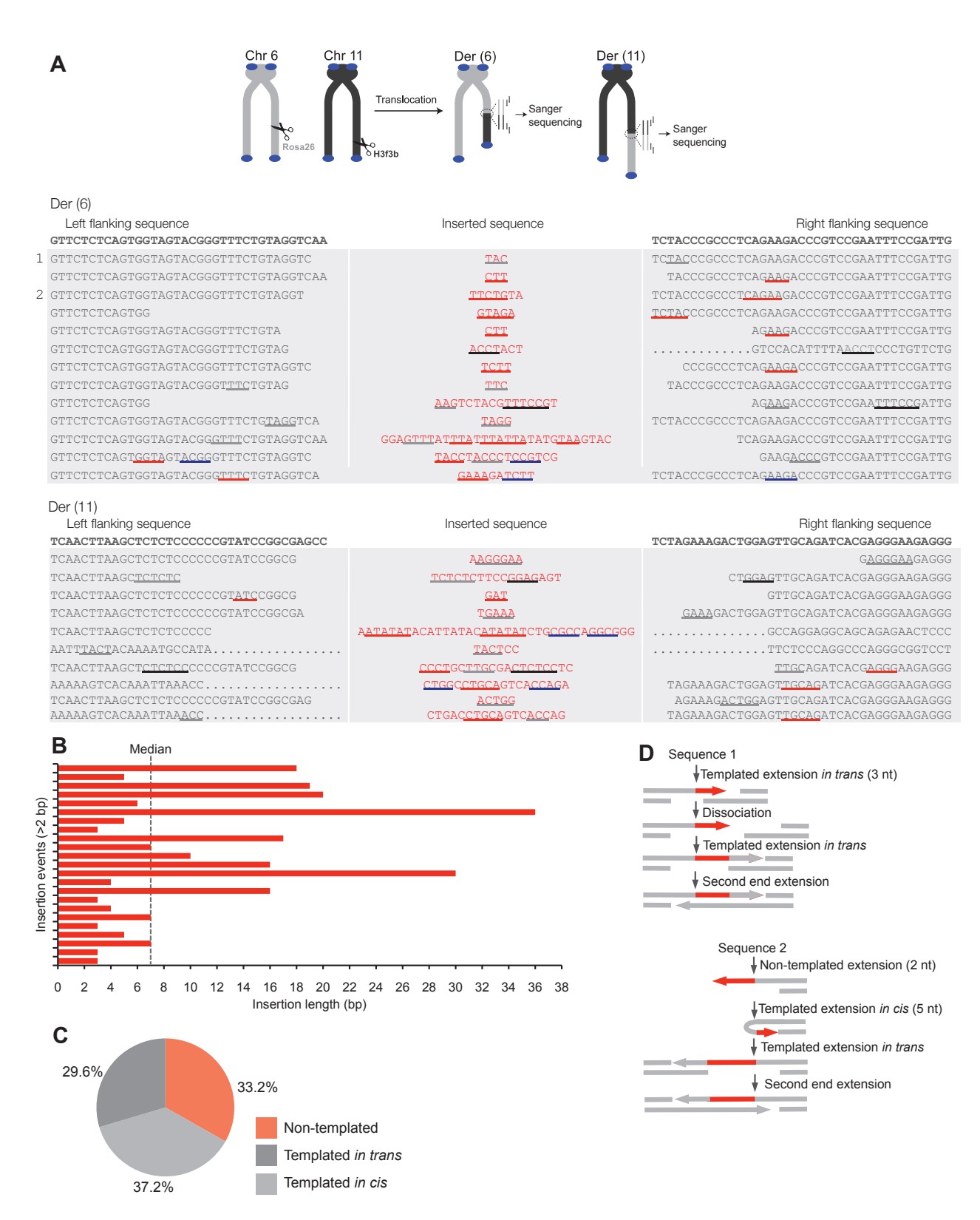

**Figure 4.** Polθ oscillates between three different modes of terminal transferase activity during alternative end-joining in vivo. (**A**) Scheme for Polθ mediated alt-EJ of site-specific DSBs in mouse embryonic stem cells (top). Sequences of alt-EJ products generated by Polθ in cells (bottom). Red text,
*Figure 4 continued on next page*

*Figure 4 continued*

insertions; black text, original DNA sequence; black and grey underlines, sequences copied from original template; red and blue underlines, complementary sequences due to snap-back replication; red sequence without underlines, random insertions; Original DNA sequences indicated at top;... , large deletions. (B) Plot of insertion tract lengths generated in panel A. (C) Chart depicting percent of individual nucleotide insertion events due to non-templated extension, templated extension *in cis* and templated extension *in trans*. t test indicates no significant difference between percent of non-templated and templated *in cis* insertions. (D) Models of Polθ activity based on end-joining products 1 and 2 from panel **A**.

The following figure supplements are available for figure 4:

**Figure supplement 1.** Large insertions copied from remote donor locations.

**Figure supplement 2.** Additional sequence analysis of alternative end-joining products generated in vivo.

(red and blue underlines) (*Figure 4A*). Individual nucleotide insertion events due to non-templated extension appear to be slightly lower in the in vivo system (33.2%) compared to the in vitro system (39%), which is likely due to a lower proportion of $Mn^{2+}$ to $Mg^{2+}$ in cells. Consistent with this, events due to templated extension *in cis* (snap-back replication) appear slightly higher in the in vivo system (37.2%) compared to the in vitro system (28.8%). We note that DNA deletions were observed in both systems, albeit more frequently in cells which is likely due to nuclease activity. Deletions in the in vitro system likely result from Polθ mediated end-joining at internal sites within the 3' overhang, as shown previously (*Kent et al., 2015*). This mechanism may also contribute to deletions observed in vivo. Regardless of the specific mechanisms underlying deletion formation in each system, the insertion tracts observed in vitro and in vivo appear similar in nature in regards to template dependency (compare *Figures 3C* and *4C*). Furthermore, the median insertion tract length (7 bp) generated by Polθ in vitro and in vivo was identical (compare *Figures 3B* and *4B*). Thus, these data demonstrate that the reconstituted alt-EJ system closely resembles the mechanism of alt-EJ in cells. We note that some large (>30 bp) insertions copied from remote chromosome sites and the CRISPR/Cas9 vector were also observed in the in vivo system (*Figure 4—figure supplement 1*). However, these insertions are likely due to a different mechanism such as strand invasion into duplex DNA. Additional analysis of end-joining products generated in vivo demonstrates that Polθ preferentially produces insertions >2 bp in length, and occasionally generates relatively long insertions (i.e. >25 bp) (*Figure 4—figure supplement 2*). Importantly, sequences of end-joining products generated in vivo support the same mechanism of Polθ switching observed in vitro (*Figure 4D*). Altogether, the results presented in *Figures 3* and *4* along with previous studies showing the requirement for Polθ in forming insertions indicate that Polθ is the main enzyme involved in generating insertions during alt-EJ. These results also indicate that Polθ oscillates between three different modes of terminal transferase activity to generate insertion mutations, and that $Mn^{2+}$ likely acts as a co-factor for Polθ in vivo.

## Polθ exhibits preferential terminal transferase activity on DNA with 3' overhangs

We next further characterized Polθ-$Mn^{2+}$ terminal transferase activity on a variety of DNA substrates. For example, we further tested Polθ-$Mn^{2+}$ on homopolymeric ssDNA composed of either deoxythymidine-monophosphates (poly-dT) or deoxycytidine-monophosphates (poly-dC), and ssDNA containing variable sequences. The polymerase preferentially extended all of the substrates by more than 100 nt in the presence of deoxyadenosine-triphosphate (dATP), regardless of the sequence context (*Figure 5A,B*). Polymerases are known to preferentially incorporate deoxyadenosine-monophosphate (dAMP) when template base coding is not available, which is referred to as the A-rule. For example, polymerases preferentially incorporate a single dAMP opposite an abasic site or at the end of a template. Thus, the observed preferential incorporation of dAMP by Polθ-$Mn^{2+}$ is consistent with the A-rule and template-independent activity. Polθ also extended ssDNA in the presence of dTTP, dCTP, and dGTP, however, the lengths of these products were shorter than with dATP (*Figure 5A,B*). For example, in the case of non-homopolymeric ssDNA, Polθ-$Mn^{2+}$ transferred ~30–70 nt in the presence of dTTP, dCTP, or dGTP (*Figure 5B*), which demonstrates that Polθ-$Mn^{2+}$ terminal transferase activity is relatively efficient even in the absence of the preferred dATP. Notably,

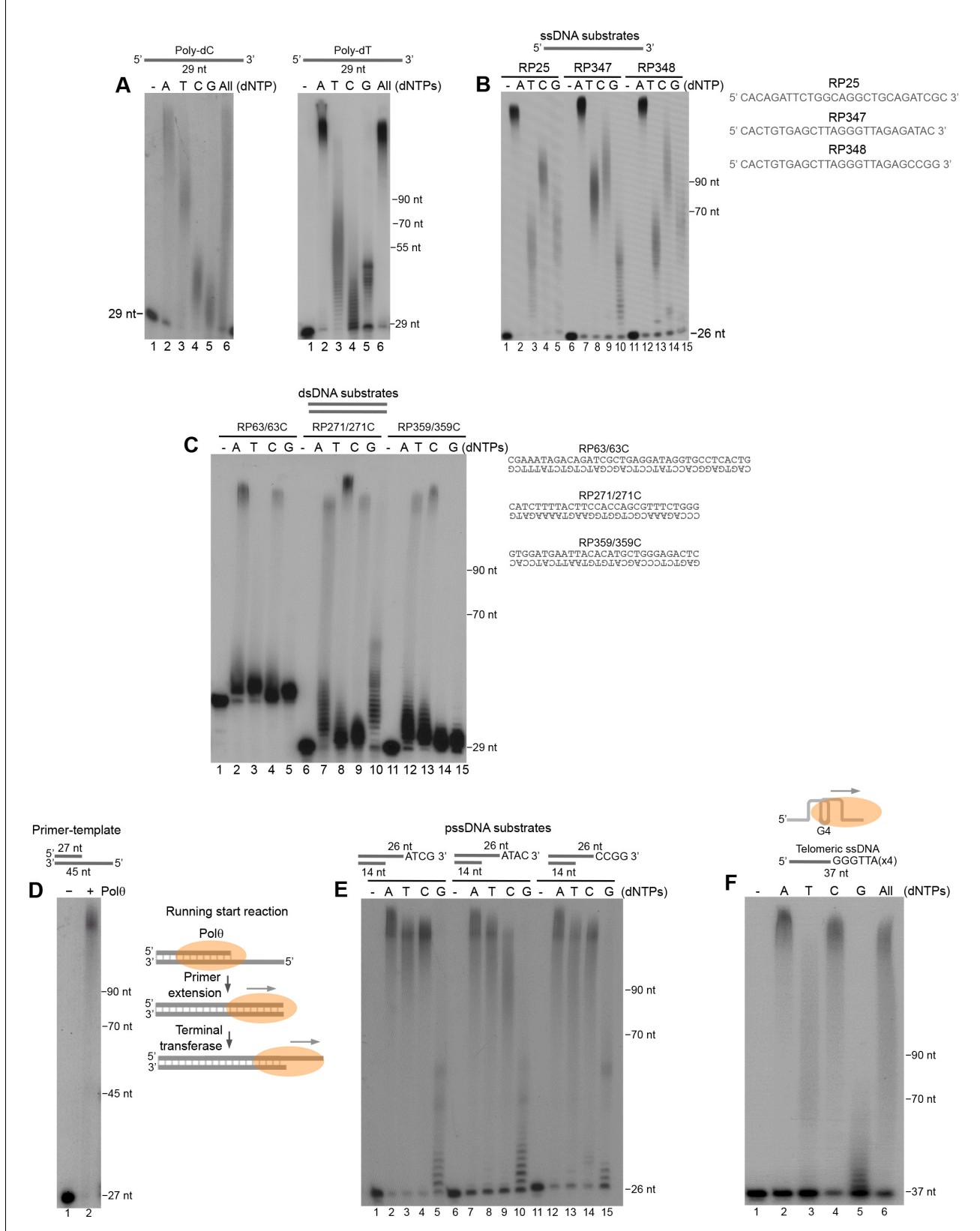

**Figure 5.** Polθ exhibits preferential terminal transferase activity on pssDNA. (**A**) Denaturing gels showing Polθ extension of poly-dC (left) and poly-dT (right) ssDNA with 5 mM $Mn^{2+}$ and the indicated dNTPs. (**B**) Denaturing gel showing Polθ extension of the indicated ssDNA with 5 mM $Mn^{2+}$ and

*Figure 5 continued on next page*

*Figure 5 continued*

indicated dNTPs. (**C**) Denaturing gel showing Polθ extension of the indicated dsDNA with 5 mM $Mn^{2+}$ and indicated dNTPs. (**D**) Denaturing gel showing Polθ extension of a primer-template with 5 mM $Mn^{2+}$ and all four dNTPs. Model of Polθ-$Mn^{2+}$ activity on a primer-template (right). (**E**) Denaturing gel showing Polθ extension of the indicated pssDNA with 5 mM $Mn^{2+}$ and indicated dNTPs. (**F**) Denaturing gels showing Polθ extension of ssDNA modeled after telomere sequence with 5 mM $Mn^{2+}$ and the indicated dNTPs.

The following figure supplement is available for figure 5:

**Figure supplement 1.** Comparison of Polθ and Polμ terminal transferase activities with $Mn^{2+}$.

the non-homologous end-joining (NHEJ) X-family polymerase, Polμ, exhibited minimal terminal transferase activity compared to Polθ under identical conditions (*Figure 5—figure supplement 1*). Previous studies similarly demonstrated limited terminal transferase activity by Polμ which is most closely related to TdT (*Andrade et al., 2009*). Thus, to date the data presented insofar indicate that, aside from TdT, Polθ possesses the most robust terminal transferase activity for the polymerase enzyme class.

We next examined the ability of Polθ-$Mn^{2+}$ to extend blunt-ended double-strand DNA (dsDNA). The results show that Polθ efficiently extends duplex DNA, however, this is limited to only 1–2 nucleotides which may be due to a lower affinity of the polymerase for blunt-ended DNA (*Figure 5C*). Interestingly, Polθ efficiently extended a primer-template far beyond the downstream end of the template (*Figure 5D*, left). Thus, the polymerase performs efficient long-range extension of dsDNA when given a running start (*Figure 5D*, right schematic).

Considering that Polθ is thought to act on DSBs partially resected by MRN and CtIP during MMEJ/alt-EJ (*Kent et al., 2015*), we examined its terminal transferase activity on pssDNA. Remarkably, Polθ-$Mn^{2+}$ exhibited the most efficient terminal transferase activity on pssDNA (*Figure 5E*). For example, the polymerase extended the pssDNA substrates to longer lengths with dTTP and dCTP, whereas dGTP was still limiting (compare *Figure 5E* with *Figure 5B*).

Consistent with its role in promoting alt-EJ of telomeres in cells deficient in telomere protection and NHEJ factors (*Mateos-Gomez et al., 2015*), we found that Polθ exhibits efficient terminal transferase activity on ssDNA modeled after telomeres which are known to contain stable G-quadruplex (G4) secondary structures (*Figure 5F*). Here again, extension in the presence of dGTP was suppressed. Considering that consecutive dGMP incorporation events limit Polθ terminal transferase activity, we presume the multiple guanosines present in telomere repeats cause a similar inhibitory effect. All other nucleotides were efficiently transferred to the telomeric ssDNA substrate (*Figure 5F*). Taken together, the results in *Figure 5* show that Polθ exhibits the most robust terminal transferase activity on pssDNA which is consistent with its role in MMEJ/alt-EJ, and that the polymerase is also efficient in extending various ssDNA substrates and dsDNA when given a running start.

## Conserved residues facilitate Pol*θ* processivity and terminal transferase activity

Next, we sought to identify structural motifs that promote Polθ terminal transferase activity. Polθ is a unique A-family polymerase since it contains three insertion loops, and previous studies have shown that loop 2 is necessary for Polθ extension of ssDNA (*Hogg et al., 2012*; *Kent et al., 2015*). The position of this motif is conserved in Polθ and is located immediately downstream from a conserved positively charged residue, arginine (R) or lysine (K), at position 2254 (*Figure 6A*). Recent structural studies of Polθ in complex with a primer-template and incoming nucleotide show that loop 2 lies relatively close to the 3' terminus of the primer, but is likely flexible in this conformation due to a lack of resolution (*Figure 6B*) (*Zahn et al., 2015*). Considering that Polθ ssDNA extension with $Mg^{2+}$ is likely related to its activity with $Mn^{2+}$, we predicted that loop 2 would also confer template-independent terminal transferase activity. Indeed, a loop 2 deletion mutant of Polθ (PolθL2) failed to extend ssDNA under optimal template-independent terminal transferase conditions with $Mn^{2+}$ (*Figure 6C*). Similar to previous results, PolθL2 fully extended a primer-template (*Figure 6D*). Here, PolθWT extension continued beyond the template due to the polymerase's robust terminal transferase activity with $Mn^{2+}$ (*Figure 6D*).

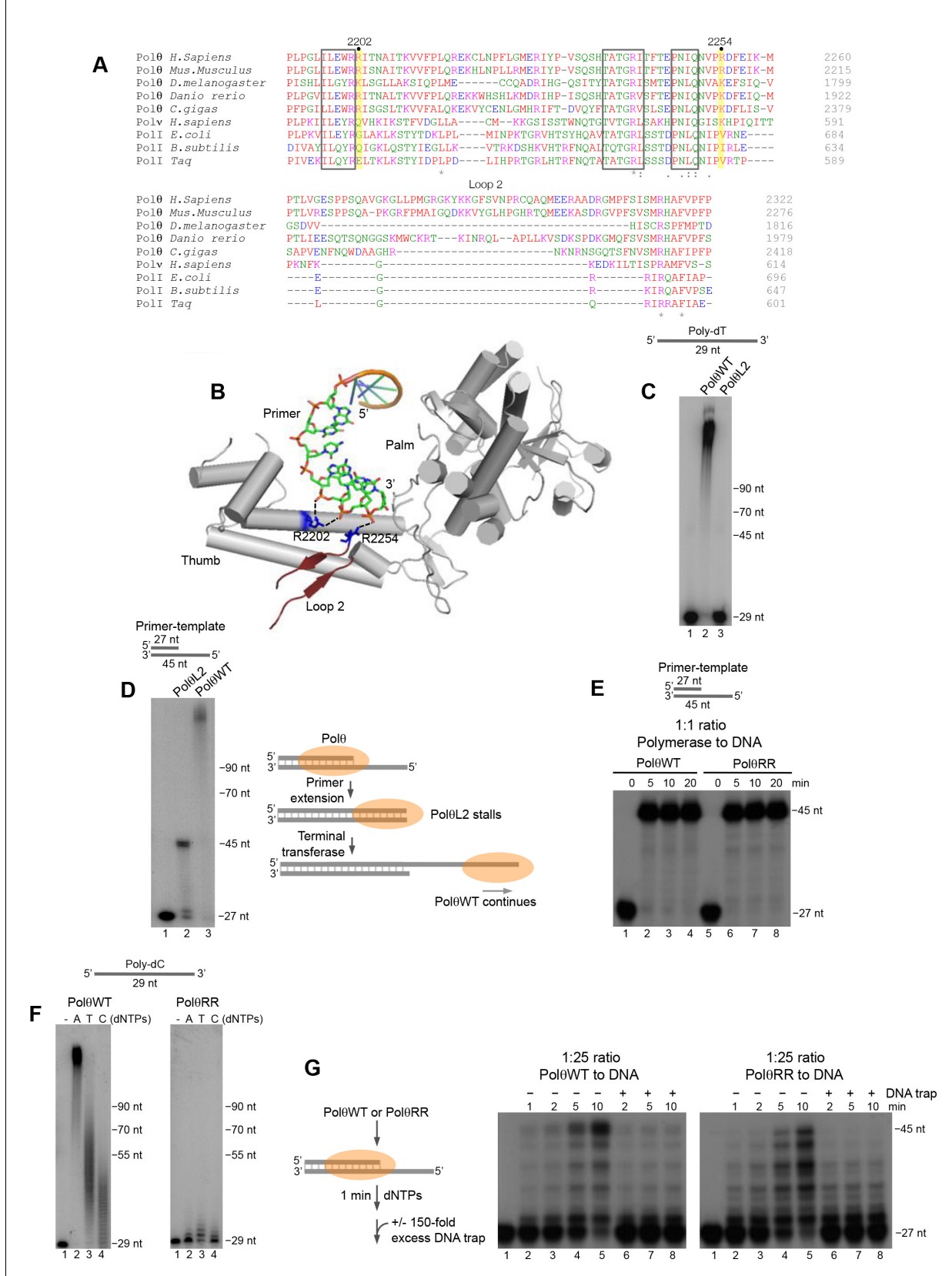

**Figure 6.** Conserved residues contribute to Polθ processivity and template-independent terminal transferase activity. (**A**) Sequence alignment of Polθ and related A-family Pols. Conserved positively charged residues (2202, 2254) and loop 2 in Polθ are highlighted in yellow and grey, respectively. Black

*Figure 6 continued on next page*

*Figure 6 continued*

boxes indicate conserved motifs. * = identical residues,: = residues sharing very similar properties,. = residues sharing some properties. Red, small and hydrophobic; Blue, acidic; Magenta, basic; Green, hydroxyl, sulfhydryl, amine. (B) Structure of Polθ with ssDNA primer (PDB code 4X0P) (*Zahn et al., 2015*). Residues R2202 and R2254 are indicated in blue. Dotted blue lines indicate ionic interactions. Loop 2 is indicated in dark red. Thumb and palm subdomains are indicated. (C) Denaturing gel showing PolθWT and PolθL2 extension of ssDNA with 5 mM $Mn^{2+}$ and all four dNTPs. (D) Denaturing gel showing PolθWT and PolθL2 extension of a primer-template with 5 mM $Mn^{2+}$ and all four dNTPs. Model of PolθWT-$Mn^{2+}$ and PolθL2-$Mn^{2+}$ activities on a primer-template (right). (E) Denaturing gel showing a time course of PolθWT and PolθRR extension of a primer-template in the presence of 10 mM $Mg^{2+}$ and all four dNTPs. (F) Denaturing gel showing PolθWT (left) and PolθRR (right) extension of poly-dC ssDNA with 5 mM $Mn^{2+}$ and the indicated dNTPs. (G) Schematic of assay (left). Denaturing gel showing PolθWT and PolθRR extension of an excess of radiolabeled primer-template with all four dNTPs and 10 mM $Mg^{2+}$ either in the presence or absence of 150-fold excess unlabeled DNA trap.

Structural studies showed that two conserved positively charged residues, R2202 and R2254, bind to the phosphate backbone of the 3' portion of the primer (*Figure 6A,B*) (*Zahn et al., 2015*). Since these positively charged residues are conserved in Polθ but not other A-family members (*Figure 6A*), we envisaged that they might contribute to Polθ terminal transferase activity. We first tested primer-extension of a double mutant version of Polθ in which R2202 and R2254 were changed to alanine (A) and valine (V), respectively (PolθRR). Recent studies showed that single R2202A and R2254V Polθ mutants were slightly defective in translesion synthesis (*Zahn et al., 2015*). PolθRR extended the primer in a similar manner to PolθWT (*Figure 6E*). Yet, PolθRR showed a severe defect in template-independent terminal transferase activity compared to PolθWT under identical conditions with $Mn^{2+}$ (*Figure 6F*). Since PolθWT performs terminal transferase activity with high processivity, we wondered whether PolθRR exhibits reduced processivity. Indeed, PolθRR showed a significant deficiency in primer extension compared to PolθWT when a large excess of DNA was present, confirming a reduction in processivity (*Figure 6G*). These data also suggest that PolθWT exhibits lower processivity during primer-template extension compared to ssDNA extension (compare *Figure 6G* and *Figure 2—figure supplement 4*). Since PolθRR is defective in processivity and template-independent terminal transferase activity, this suggests that the polymerase must be processive on ssDNA to effectively perform template-independent terminal transferase activity. Together, these data identify conserved residues that contribute to Polθ terminal transferase activity by conferring processivity onto the enzyme through binding the 3' primer terminus.

## Comparison of Pol*θ* and TdT terminal transferase activities

Importantly, terminal transferase activity is widely used to modify ssDNA ends for various types of applications including biotechnology, biomedical research, and synthetic biology. Currently, the only enzyme developed and marketed for these applications is terminal deoxynucleotidyl transferase (TdT) whose cellular function is to promote antibody diversity by transferring non-templated nucleotides to V, D and J exon regions during antibody gene maturation (*Motea and Berdis, 2010*). We compared the activities of Polθ and TdT in *Figure 7A*. Remarkably, Polθ exhibited a similar ability to extend ssDNA as TdT assayed under optimal conditions recommended by the supplier (*Figure 7A*). The results also show that in this reaction Polθ and TdT preferentially utilize dATP and dTTP, respectively, which suggests different mechanisms of action (*Figure 7A*).

Many biotechnology and biomedical research applications require ssDNA substrates modified with fluorophores or other chemical groups, such as those that enable DNA attachment to solid surfaces. We therefore examined the ability of Polθ to transfer deoxyribonucleotides and ribonucleotides conjugated with different functional groups to the 3' terminus of ssDNA. Again, using the supplier's recommended assay conditions for TdT, and identical concentrations of Polθ under its optimal conditions, we unexpectedly found that Polθ-$Mn^{2+}$ is more effective in transferring ribonucleotides to ssDNA compared to TdT (*Figure 7B*). Although previous studies have shown that Polθ strongly discriminates against ribonucleotides (*Hogg et al., 2012*), this fidelity mechanism is largely compromised under our conditions used for terminal transferase activity. Again, using the respective optimal conditions for Polθ and TdT at identical concentrations, we also found that Polθ-$Mn^{2+}$ is more proficient in transferring most modified deoxy-ribonucleotides and ribonucleotides to ssDNA than TdT (*Figure 7C,D*). For example, Polθ more efficiently transferred eight out of ten modified nucleotides tested. In some cases, Polθ produced longer extension products than TdT (*Figure 7C*). In other cases, Polθ transferred nucleotides that TdT was unable to incorporate (*Figure 7C*, black

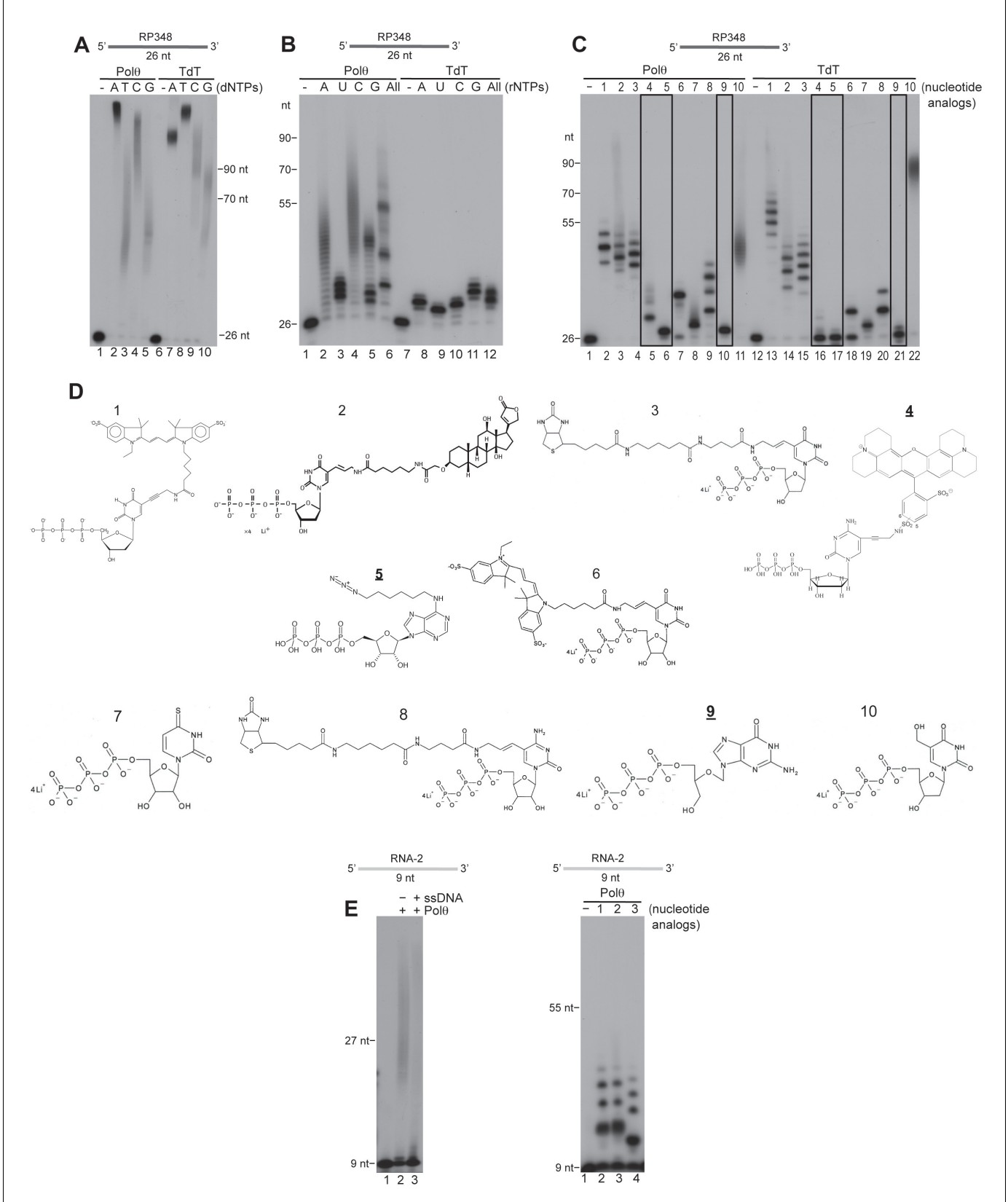

**Figure 7.** Comparison of Polθ and TdT terminal transferase activities. (**A**) Denaturing gel showing Polθ-Mn$^{2+}$ (lanes 1–5) and TdT (lanes 6–10) extension of ssDNA with the indicated dNTPs. (**B**) Denaturing gel showing Polθ-Mn$^{2+}$ (lanes 1–6) and TdT (lanes 7–12) extension of ssDNA with the indicated

*Figure 7 continued on next page*

*Figure 7 continued*

ribonucleotides (rNTPs). (**C**) Denaturing gel showing Polθ-Mn$^{2+}$ (lanes 1–11) and TdT (lanes 12–22) extension of ssDNA with the indicated nucleotide analogs illustrated in panel (d). Boxed lanes indicate nucleotides analogs that are exclusively transferred by Polθ-Mn$^{2+}$. (**D**) Nucleotide analogs: 1, cy3-dUTP; 2, Digoxigenin-11-dUTP; 3, Biotin-16AA-dUTP; 4, Texas Red-5-dCTP; 5, N6 -(6-Azido)hexyl-ATP; 6, Cyanine 3-AA-UTP; 7, 4-Thio-UTP; 8, Biotin-16AA-CTP; 9, Ganciclovir Triphosphate; 10, 5-Hydroxymethyl-2′-deoxyuridine-5′-Triphosphate. Underlined nucleotide analogs (4,5,9) are exclusively transferred by Polθ-Mn$^{2+}$. (**E**) Denaturing gel showing Polθ-Mn$^{2+}$ extension of RNA with all four dNTPs in the presence (lane 3) and absence (lane 2) of unlabeled ssDNA (left panel). Denaturing gel showing Polθ-Mn$^{2+}$ extension of RNA with the indicated nucleotide analogs (right panel). Polθ-Mn$^{2+}$ extension assays (**A-C,E**) included 5 mM Mn$^{2+}$.

boxes). For instance, Polθ efficiently transferred a nucleotide containing a linker attached to an azide group which is widely used for "click chemistry" applications (*Figure 7C*, lane 6). In contrast, TdT failed to transfer this nucleotide altogether (*Figure 7C*, lane 17). Moreover, TdT failed to transfer nucleotides containing a modified sugar and a linker attached to Texas Red, whereas these substrates were efficiently incorporated by Polθ (*Figure 7C*, nucleotide analogs 4 and 9). These results show that Polθ efficiently transfers ribonucleotides and deoxyribonucleotides containing modifications on their base moieties, such as fluorophores and functional groups including biotin and digoxigenin, as well as nucleotides containing sugar modifications (i.e. ganciclovir mono-phosphate). Considering that Polθ also exhibits translesion synthesis activity, these results may be attributed to its natural ability to accommodate non-canonical nucleotides in its active site (*Hogg et al., 2011*; *Yoon et al., 2014*).

Lastly, we investigated whether Polθ exhibits terminal transferase activity on RNA. Surprisingly, Polθ transferred both canonical and modified nucleotides to RNA, albeit less efficiently than to DNA (*Figure 7E*). Together, the results presented in *Figure 7* characterize Polθ as among the most proficient terminal transferases identified and demonstrate that Polθ is more effective than TdT in modifying nucleic-acid substrates for biomedical research and biotechnology applications.

## Discussion

Recent studies have discovered that mammalian Polθ is essential for MMEJ/alt-NHEJ, which promotes chromosome rearrangements and resistance to DNA damaging agents, including those used for chemotherapy (*Kent et al., 2015*; *Mateos-Gomez et al., 2015*; *Yousefzadeh et al., 2014*). Polθ was previously shown to be essential for alt-EJ in flies and worms (*Chan et al., 2010*; *Koole et al., 2014*), demonstrating a conserved role for this polymerase in higher eukaryotes. These cellular studies have shown that two types of insertions, non-templated and templated, are generated at alt-EJ repair junctions which are dependent on Polθ expression (*Chan et al., 2010*; *Koole et al., 2014*; *Mateos-Gomez et al., 2015*; *Yousefzadeh et al., 2014*). In the case of non-templated insertions, it has been proposed that Pol promotes random transfer of nucleotides via a putative template-independent terminal transferase activity (*Mateos-Gomez et al., 2015*). Yet, biochemical studies have shown that Polθ lacks template-independent terminal transferase activity, creating a paradox between cellular and in vitro data (*Kent et al., 2015*; *Yousefzadeh et al., 2014*). In the case of templated insertions, a copy *in trans* model has been proposed which also has not been proven in vitro (*Chan et al., 2010*; *Koole et al., 2014*; *Yousefzadeh et al., 2014*). In this report, we elucidate how Polθ generates both templated and non-templated nucleotide insertion mutations during alt-EJ, and characterize the polymerase as a highly robust terminal transferase for biotechnology and biomedical research applications.

We first discover that Polθ exhibits robust template-independent terminal transferase activity in the presence of Mn$^{2+}$. Considering that structural studies show that differential binding of divalent cations within the active site of Polθ slightly alters its local conformation (*Zahn et al., 2015*), Mn$^{2+}$ binding likely facilitates an active site conformation more favorable for non-templated DNA synthesis. Since Polθ dependent non-templated nucleotide insertions are commonly associated with alt-EJ in cells, our findings suggest that Mn$^{2+}$ acts as a co-factor of Polθ in vivo. For example, although the concentration of Mn$^{2+}$ is relatively low in cells (~0.2 mM) and is considerably less than Mg$^{2+}$ (~1.0 mM), we show that these concentrations of Mn$^{2+}$ and Mg$^{2+}$ stimulate Polθ template-independent terminal transferase activity by 3–8 fold. Thus, cellular concentrations of Mn$^{2+}$ are likely to activate Polθ template-independent activity. Intriguingly, Mn$^{2+}$ has been shown to act as a necessary

co-factor for the MRX nuclease complex and its mammalian counterpart, MRN, which is also essential for alt-EJ due to its role in generating 3' ssDNA overhangs onto which Polθ acts (*Cannavo and Cejka, 2014*; *Trujillo et al., 1998*). Thus, various enzymes involved in DNA repair are likely to utilize $Mn^{2+}$ as a co-factor in addition to $Mg^{2+}$.

To our surprise, the Polθ-$Mn^{2+}$ complex exhibited a higher efficiency of transferring ribonucleotides and most modified nucleotide analogs to the 3' terminus of ssDNA than TdT at identical concentrations. For example, in the presence of ribonucleotides, Polθ-$Mn^{2+}$ generated substantially longer extension products, which demonstrates a lower discrimination against ribonucleotides. Polθ-$Mn^{2+}$ also produced longer extension products than TdT in the presence of most nucleotide analogs, including those that contain large functional groups. Moreover, Polθ-$Mn^{2+}$ efficiently transfered certain nucleotide analags that TdT failed to utilize as substrates. For instance, we found that Polθ-$Mn^{2+}$ exclusively transfered a nucleotide conjugated with Texas Red and a nucleotide containing an azide group which is widely used for 'click' chemistry applications. We additionally found that Polθ-$Mn^{2+}$ is capable of transferring canonical and modified nucleotides to RNA, albeit with lower efficiency than DNA. Based on these unexpected findings, we anticipate that Polθ will be more useful for modifying nucleic acid substrates for biotechnology, biomedical research and synthetic biology applications. Moreover, since Polθ does not require toxic reaction components like TdT, such as $Co^{2+}$ salts or salts of cacodylic acid, Polθ terminal transferase assays are a safer option for research and biotechnology applications.

Our report raises the question why evolution selected for two robust terminal transferases: Polθ and TdT. It is well known that the primary function of TdT is to generate insertion mutations during NHEJ of V, D and J antibody gene regions, which promotes antibody diversity that is necessary for a strong immune system (*Motea and Berdis, 2010*). Since a diverse immunological defense is important for survival, a clear selective pressure for TdT existed. In the case of Polθ, it appears that the polymerase has also been selected to generate insertion mutations during end-joining, however, the evolutionary pressure for this particular mechanism is not as clear. For example, although Polθ is essential for alt-EJ, this pathway appears to occur infrequently compared to primary DSB repair processes, such as HR (*Mateos-Gomez et al., 2015*; *Truong et al., 2013*). Consistent with this, Polθ is not important for normal cell survival or development. Recent studies of *C. elegans*, however, surprisingly show that Polθ mediated alt-EJ is a primary form of repair in germ cells (*van Schendel et al., 2015*). Furthermore, it was shown that Polθ mediated alt-EJ promotes a deletion and insertion (indel) signature in propogated laboratory strains that is similar to indels found in natural isolates (*van Schendel et al., 2015*). These studies therefore suggest that Polθ is important for generating genetic diversity. Interestingly, human Polθ is highly expressed in testis, suggesting the polymerase might also play a role in facilitating genetic diversity in mammals (*Seki et al., 2003*).

Considering that alt-EJ also promotes replication repair as a backup to HR, Polθ likely benefits cell survival at the expense of indels when lethal DSBs fail to be repaired by the primary HR pathway (*Truong et al., 2013*). For example, Polθ mediated alt-EJ in *C. elegans* was shown to facilitate replication repair at stable G4 structures which may pose problems for the HR machinery and therefore potentially require an alternative and more accommodating error-pone form of repair (*Koole et al., 2014*). Polθ has also been shown to suppress large genetic deletions in *C. elegans*, which demonstrates an obvious benefit for the polymerase (*Koole et al., 2014*). Yet, whether these various functions of Polθ are conserved in mammals awaits further research.

Our studies reveal that Polθ generates nucleotide insertions by oscillating between multiple mechanisms, which portrays a promiscuous enzyme that readily extends ssDNA by almost any means in order to catalyze end-joining products that frequently contain insertion mutations. For example, we observed that Polθ generates nucleotide insertions during alt-EJ in vitro by spontaneously switching between three distinct modes of terminal transferase activity: non-templated extension, templated extension *in cis*, and templated extension *in trans*. Importantly, we show that the characteristics of these insertions are nearly identical to those generated by Polθ mediated alt-EJ in cells, which indicates that Polθ also switches between these three mechanisms of terminal transferase activity in vivo. To our knowledge, the ability of a polymerase to spontaneously switch between three distinct modes of DNA synthesis has not been demonstrated. Thus, our data reveal an unprecedented set of mechanisms by which a single polymerase can synthesize DNA, presumably for generating genetic diversity and as a last resort for repairing lethal DSBs at the expense of mutations.

## Materials and methods

### Polθ terminal transferase activity

500 nM Polθ was incubated with 50 nM of the indicated 5' $^{32}$P-labeled DNA for 120 min at 42°C (or other indicated time intervals and temp) in the presence of 0.5 mM of indicated dNTPs in a 10 µl volume of buffer A (20 mM TrisHCl pH 8.2, 10% glycerol, 0.01% NP-40, 0.1 mg/ml BSA) with indicated divalent cations; optimal Polθ terminal transferase activity was performed with 5 mM MnCl$_2$. Reactions were terminated by the addition of 20 mM EDTA and 45% formamide and DNA was resolved by electrophoresis in urea polyacrylamide gels then visualized by autoradiography. Polm terminal transferase reactions were performed using the same conditions as Polθ. 50 nM Polθ was used in experiments employing ssDNA traps. 150-fold excess of unlabled ssDNA trap was added to reactions at indicated time points where indicated. Polθ terminal transferase activity in solid-phase. 50 nM RP347B was immobilized to magnetic streptavidin beads (Dynabeads M-270, Invitrogen) in buffer A supplemented with 100 mM NaCl. Excess unbound DNA was then removed by washing beads 3x with buffer A with 100 mM NaCl. Next, the bead-DNA mixture was washed and resuspended in buffer A containing 10 mM MgCl and 1 mM MnCl. 500 nM Polθ was then added for 10 min to allow for ssDNA binding. Excess unbound Polθ was then removed by washing the beads 4x with 200 µl buffer A supplemented with 10 mM MgCl and 1 mM MnCl. Beads were resuspended in buffer A supplemented with 10 mM MgCl and 1 mM MnCl, then 0.5 mM dNTPs were added at 42°C. After 15 s, either dH$_2$0 or 7.5 µM RP427 was added and the reaction was terminated after 120 min by addition of EDTA. The beads were thoroughly washed to remove excess ssDNA trap. The beads were then resuspended in dH$_2$0 followed by boiling for 1–2 min. The supernatant was collected, then another cycle of boiling and supernatant collection was performed. The DNA from the supernatant was purified using Zymo DNA Clean and Concentrator-5 kit. Purified DNA was then ligated to RP430P overnight at room temp using T4 RNA ligase (New Englan Biolabs). RNA ligase was denatured at 65°C, then the DNA was purified using Zymo DNA Clean and Concentrator-5 kit. The ligated DNA was then amplified via PCR using GoTaq Green (Promega) and primers RP347 and RP431. PCR products were purified using QIAquick PCR purification kit (Qiagen). Pure PCR products were then cloned into E. coli plasmid vectors using TOPO TA cloning (Invitrogen). Individual plasmids containing PCR products were amplified in E. coli, isolated, then sequenced.

### Polθ mediated alt-EJ in vitro

Equimolar concentrations (100 nM) of pssDNA substrates RP429/RP430-P and RP434-P/RP408 were mixed with 50 nM Polθ and 88.5 nM Lig3 in buffer A supplemented with 1 mM MnCl$_2$, 10 mM MgCl$_2$ and 1mM ATP. Next, 10 µM dNTPs were added for 120 min at 37°C in a total volume of 100 µl. Reactions were terminated by incubation at 80°C for 20 min. (Negative control reactions included: omission of Lig3, and; omission of Polθ and Lig3). DNA was purified using QIAquick Nucleotide Removal kit (QIAGEN) then amplified using PCR Master Mix (Promega) and end-joining specific primers RP431 and RP435. PCR products were purified using GeneJET PCR Purification Kit (ThermoScientific) then cloned into the pCR2.1-TOPOvector (Invitrogen). DNA was transformed into *E. coli* DH5α cells, and individual plasmids from single colonies were purified and sequenced. Polθ mediated alt-EJ in *Figure 3—figure supplement 3* was performed as described above, however, 1 mM MgCl$_2$, 50 µM MnCl$_2$ and 100 µM dNTPs were used. Where indicated, 150-fold excess (15 µM) of ssDNA trap (RP347) was added to the reaction at the indicated time point.

### Polθ mediated alt-EJ in cells

Polθ mediated alt-EJ involving chromosomal translocation was performed as previously described (*Mateos-Gomez et al., 2015*). Briefly, mouse Embryonic Stem (ES) cells were transfected with 3 µg of Cas9-gRNA(Rosa26;H3f3b) (*Mateos-Gomez et al., 2015*). After transfection, $5 \times 10^4$ cells were seeded per well in a 96-well plate, and lysed 3 days later in 40 µl lysis buffer (10 mM Tris pH 8.0, 0.45% Nonidet P-40, 0.45% Tween 20). The lysate was incubated with 200 µg/ml of Proteinase K for 2 hr at 55°C. Translocation detection was performed using nested PCR. The primers used in the first PCR reaction include Tr6-11-Fwd:5'-GCGGGAGAAATGGATATGAA-3'; Tr6-11-Rev: 5'-TTGACGCCTTCCTTCTTCTG -3', and Tr11-6-Fwd: 5'-AACCTTTGAAAAAGCCCACA-3' and Tr11-6-Rev:5'-GCACGTTTCCGACTTGAGTT-3', for Der(6) and Der (11) respectively. For the second round

of PCR amplification, the following primers were used: Tr6-11NFwd: 5′-GGCGGATCACAAGCAA TAAT-3′; Tr6-11NRev: 5′-CTGCCATTCCAGAGATTGGT-3′ and Tr11-6NFwd:5′-AGCCACAGTGC TCACATCAC-3′ and Tr11-6NRev:5′TCCCAAAGTCGCTCTGAGTT-3′. Amplified products corresponding to translocation events were subject to Sanger sequencing to determine the junction sequences.

## TdT terminal transferase activity

TdT terminal transferase reactions were performed on indicated 5′ $^{32}$P-labeled DNA using conditions recommended by New England Biolabs: 50 mM potassium acetate, 20 mM Tris acetate, 10 mM magnesium acetate, pH 7.9, with 0.25 mM cobalt and incubated at 37°C. Incubation times and DNA concentrations were identical as experiments with Polθ. TdT was either used at concentrations recommended by New England Biolabs (0.2 units/μl) or equimolar concentrations as Polθ as indicated in text. DNA products were resolved as indicated above.

## Pol$\theta$ extension of RP347 and preparation of DNA for sequencing

Polθ (500 nM) was incubated with 50 nM RP347 ssDNA along with 0.5 mM dNTPs in 100 μl of buffer A supplemented with either 5 mM MnCl$_2$ or 1 mM MnCl$_2$ and 10 mM MgCl$_2$ for 120 min at 42°C. Reactions were terminated by the addition of 25 μl of 5X non-denaturing stop buffer (0.5 M Tris-HCl, pH 7.5, 10 mg/ml proteinase K, 80 mM EDTA, and 1.5% SDS). This was followed by phenol-chlorophorm extraction, ethanol precipitation, then ligation to 5′-phosphorylated RP359-P ssDNA using T4 RNA ligase (NEB). DNA products were ethanol precipitated then dissolved in water. Next, PCR amplification of ligation products was performed using primers RP347 and RP359C and Taq Master Mix (Promega). PCR products were purified using GeneJET PCR Purification Kit (Thermo-Scientific) then cloned into the pCR2.1-TOPO vector (Invitrogen). DNA was transformed into *E. coli* DH5α cells, and individual plasmids from single colonies were purified and sequenced.

## Pol$\theta$-Mg$^{2+}$ primer-template extension

Polθ-Mg$^{2+}$ primer-extension was performed as described (*Kent et al., 2015*) with either 10 mM MgCl or 5 mM MnCl and indicated dNTPs and time intervals. Primer-extension in solid-phase was performed as follows. A 2:1 ratio of template (RP409) to biotinylated primer (RP25B) was annealed then immobilized to magnetic streptavidin beads (Dynabeads M-270, Invitrogen) pre-washed with buffer A supplemented with 100 mM NaCl. Excess unbound DNA was then removed by washing beads 3x with 200 μl of buffer A with 100 mM NaCl. Next, the bead-DNA mixture was washed and resuspended in buffer A containing 5 mM MnCl and 0.5 mM dNTPs. 500 μM Pol was then added for 120 min at 42°C. The reaction was then terminated by the addition of 20 mM EDTA followed by boiling for 1–2 min. The supernatant was collected, then another cycle of boiling and supernatant collection was performed. The DNA from the supernatant was purified using Zymo DNA Clean and Concentrator-5 kit. Purified DNA was then ligated to RP430P overnight at room temp using T4 RNA ligase (New Englan Biolabs). RNA ligase was denatured at 65°C, then the DNA was purified using Zymo DNA Clean and Concentrator-5 kit. The ligated DNA was then amplified via PCR using GoTaq Green (Promega) and primers RP25 and RP431. PCR products were purified using QIAquick PCR purification kit (Qiagen). Pure PCR products were then cloned into E. coli plasmid vectors using TOPO TA cloning (Invitrogen). Individual plasmids containing PCR products were amplified in E. coli, isolated, then sequenced. Where indicated primer-extension was performed with either a 1:1 ratio of PolθWT or PolθRR to primer-template (50 nM), or a 1:25 ratio of PolθWT or PolθRR to primer-template (50 nM). A 150-fold excess of ssDNA trap (7.5 μM RP316) was added 1 min after initiation of primer-extension where indicated.

## De novo nucleic acid synthesis

500 nM Polθ was incubated with the indicated nucleotides at the following concentrations (500 nM ATP,UTP,GTP,dATP,dTTP,dGTP; 97 nM dCTP, [α-32P]- 6000 Ci/mmol 20 mCi/ml(Perkin Elmer)) for the indicated time intervals at 42°C in buffer A supplemented with 5 mM MnCl. Nucleic acid products were resolved in denturing polyacrylamide gels and visualized by autoradiography.

## Proteins

PolθWT and mutant proteins PolθL2 and PolθRR were purified as described (*Kent et al., 2015*). Site-directed mutagenesis was performed using QuickChange II Site-Directed Mutagenesis Kit (Agilent Technologies, Santa Clara, CA). TdT was purchased from New England Biolabs (NEB). Polμ and Lig3 were purchased from Enzymax.

## DNA

pssDNA, dsDNA and primer-templates were assembled by mixing equimolar concentrations of ssDNA substrates together in deionized water, then heating to 95–100°C followed by slow cooling to room temp. ssDNA was 5' $^{32}$P-labeled using $^{32}$P-γ-ATP (Perkin Elmer) and T4 polynucleotide kinase (NEB).

DNA (Integrated DNA technologies (IDT)) and RNA (Dharmacon) oligonucleotides (5'-3').
RP25: CACAGATTCTGGCAGGCTGCAGATCGC
RP25B: Biotin-CACAGATTCTGGCAGGCTGCAGATCGC
RP347: CACTGTGAGCTTAGGGTTAGAGATAC
RP348: CACTGTGAGCTTAGGGTTAGAGCCGG
RP63: CGAAATAGACAGATCGCTGAGGATAGGTGCCTCACTG
RP63C: CAGTGAGGCACCTATCCTCAGCGATCTGTCTATTTCG
RP271: CATCTTTTACTTCCACCAGCGTTTCTGGG
RP271C: CCCAGAAACGCTGGTGGAAGTAAAAGATG
RP359: GTGGATGAATTACACATGCTGGGAGACTC
RP359C: GAGTCTCCCAGCATGTGTAATTCATCCAC
RP266: TTTTTTTTTTTTTTTTTTGCGATCTGCAGCCTGCCAGAATCTGTG
RP331: ACTGTGAGCTTAGGGTTAGGGTTAGGGTTAGGGTTAG
RP340: CACTGTGAGCTTAGGGTTAGAGATCG
RNA-2: AUCGAGAGG
RP343-P: /5Phos/CTAAGCTCACAGTG
RP429: GGAGGTTAGGCACTGTGAGCTTAGGGTTAGAGATAC
RP430-P: /5Phos/CTAAGCTCACAGTGCCTAACCTCC
RP434-P: /5Phos/GAGCACGTCCAGGCGATCTGCAGCCTG
RP408: GAGCACGTCCAGGCGATCTGCAGCCTGCCAGAATCTGTG
RP427: CGCCACCTCTGACTTGAGCG
RP409: GAGCACGTCCACGCGATCTGCAGCCTGCCAGAATCTGTG
RP347B: Biotin-CACTGTGAGCTTAGGGTTAGAGATAC

pssDNA substrates: RP347/RP343-P, RP348/RP343-P, RP340/RP343-P, RP429/RP430-P, RP434-P/RP408. Telomeric ssDNA, RP331. Primer-templates, RP25/RP266, RP25/409, RP25B/409.

## Nucleotide analogs

1, cy3-dUTP (Santa Cruz Biotech.); 2, Digoxigenin-11-dUTP (Sigma); 3, Biotin-16AA-dUTP (TriLink Biotech.); 4, Texas Red-5-dCTP (PerkinElmer); 5, N6 -(6-Azido)hexyl-ATP (Jena Bioscience); 6, Cyanine 3-AA-UTP (TriLink Biotech.); 7, 4-Thio-UTP (TriLink Biotech.); 8, Biotin-16-AACTP (TriLink Biotech.); 9, Ganciclovir Triphosphate (TriLink Biotech.); 10, 5-Hydroxymethyl-2'-deoxyuridine-5'-Triphosphate (TriLink Biotech.).

## Acknowledgements

Research was funded by National Institutes of Health grant 1R01GM115472-01 and in part by grant 4R00CA160648-03 awarded to RTP, and Temple University School of Medicine start-up funds to RTP. We thank S Wallace (University of Vermont) for PolθWT and PolθL2 expression vectors.

## Additional information

### Competing interests

TK, RTP: Filed a provisional patent application about the use of DNA polymerase theta to modify the 3' terminus of nucleic acids. The other authors declare that no competing interests exist.

### Funding

| Funder | Grant reference number | Author |
|---|---|---|
| National Institute of General Medical Sciences | 1R01GM115472-01 | Richard T Pomerantz |
| National Institutes of Health | 4R00CA160648-03 | Richard T Pomerantz |

The funders had no role in study design, data collection and interpretation, or the decision to submit the work for publication.

### Author contributions

TK, Designed, performed and interpreted experiments, Conception and design, Acquisition of data, Analysis and interpretation of data; PAM-G, Responsible for cellular alt-EJ experiments and provided editorial input, Acquisition of data, Analysis and interpretation of data; AS, Responsible for cellular alt-EJ experiments and provided editorial input, Acquisition of data, Analysis and interpretation of data, Drafting or revising the article; RTP, Designed, performed and interpreted experiments, Conceived the study, Wrote the manuscript, Conception and design, Acquisition of data, Analysis and interpretation of data, Drafting or revising the article

### Author ORCIDs

Richard T Pomerantz, http://orcid.org/0000-0003-1194-9871

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
