## [Decision Letter]

Thank you for submitting your work entitled "Polymerase θ is a Robust Terminal Transferase that Oscillates Between Three Different Mechanisms During End-Joining" for consideration by *eLife*. Your article has been reviewed by three peer reviewers, one of whom is a member of our Board of Reviewing Editors, and the evaluation has been overseen by John Kuriyan as the Senior Editor.

The reviewers have discussed the reviews with one another and the Reviewing Editor has drafted this decision to help you prepare a revised submission.

Summary:

This paper investigates the mechanism of micro homology mediated ("alternative ") end joining by DNA polymerase theta. Genetic analysis previously showed that pol theta is required for templated and non-tempated insertions at alt-EJ junctions, and this group had previously shown that pol theta could promote DNA synapse formation of 3' ssDNA overhangs containing a minimal amount ({greater than or equal to}2 base pairs (bp)) of sequence microhomology, and then use the opposing ssDNA overhang as a template in trans to extend the DNA. However, the pol θ dependent, non-templated products were not accounted for by pol θ's in vitro activities. In this paper, Pomerantz and colleagues show that in the presence of Mn^2+^, pol θ exhibits robust, template-independent terminal deoxynucleotidyl transferase activity. It also exhibits two other synthesis modes (template directed synthesis in cis, also called snap-back synthesis, and template extension in trans). By sequencing pol θ products, they show that single DNA molecules are generated by all three of the above DNA synthesis modes. They also show evidence that similar products exist in vivo. They propose that pol theta binds to DNA ends and then cycles between the three synthesis modes until end joining is achieved. They also characterize the terminal transferase activity of pol theta and show that it is more robust than that of TdT, especially in its ability to add a wide range of bulky nucleotides to DNA ends.

This paper provides a potentially important contribution to the elucidation of the mechanism of alt-EJ. The idea that pol θ can carry out DNA synthesis in three different modes is novel and interesting. The terminal transferase activity of pol theta may very well become an important biotechnology tool.

Essential revisions:

For the most part, the data are performed to a high standard. The following, major issues must be addressed.

1) Mutagenesis due to Mn^2+^ has been extensively documented for many polymerases of different families. This metal ion is known to interfere with polymerase fidelity as these enzymes become more proficient at making mismatches but also discriminate less against the wrong sugar moiety. Mn^2+^ is also known to activate the elongation and TdT activity of pol u. RNA polymerases are also affected: poly(A) polymerase can synthesize a poly(A) oligomer with just ATP and Mn^2+^, whereas in the presence of magnesium it necessitates an RNA primer. So it should not come as a surprise that pol θ is more efficient at incorporating ribonucleotides and modified nucleotides in the presence of Mn^2+^. The authors make a statement that the switching mechanism facilitated by Mn^2+^ is used to generate diversity during alternative end-joining in vivo. Going from the experiments reported in this manuscript to saying that Mn^2+^ is a cofactor in vivo requires a huge leap of faith. The concentrations of Mn^2+^ used in the in vitro experiments are well above the reported Mn^2+^ concentration in cellulo: 1 to 5 mM vs. 10-200 uM. Moreover, gel 1E shows that at 10 mM Mg^2+^ (which is in the range of Mg^2+^ concentration in the nucleus: 0.5- 10 mM) the stimulatory effect of 1 mM Mn^2+^ is essentially non existent. It is important to show that low μM amounts of Mn^2+^ support the claimed activities. Mn^2+^ is notorious in making the DNA synthesis reaction promiscuous, and given that Pol theta has a very low fidelity (Kunkel's work), the reaction can become even more nonspecific with Mn(II). It is not ruled out that the template independent activity is not misincorporations or slippages.

2) The authors very strongly imply that one and the same pol θ molecule (or dimer) cycles through the different modes of synthesis. Their data does not justify this conclusion. They need to repeat their in vitro end-joining analysis with sequencing under conditions where dissociation and rebinding are not allowed.

[Editors' note: further revisions were requested prior to acceptance, as described below.]

Thank you for resubmitting your work entitled "Polymerase θ is a Robust Terminal Transferase that Oscillates Between Three Different Mechanisms to Generate Genetic Diversity During Alternative End-Joining" for further consideration at *eLife*. Your revised article has been favorably evaluated by John Kuriyan (Senior editor), and a Reviewing editor. The manuscript has been greatly improved but there are some remaining issues that need to be addressed before acceptance, as outlined below:

1) Figure 6 seems to argue that even WT pol θ is not highly processive in the primer-template extension reaction, as addition of the DNA trap dramatically reduces observed extension. This contrasts with the reported processivity in terminal transferase activity. The authors should note and discuss this difference in the text. Moreover, it is unclear why primer extension appears to stop at 45 nt in Figure 6 but proceeds to >100 nt in Figure 5 and Figure 6. Perhaps this is due to different reaction times, but these times are not noted in the figure itself or legend for 5D and 6D.

2) No statistical tests are presented.

3) The authors state "We note that Mn^2+^ had a greater stimulatory effect in the absence of Mg^2+^, which shows competition between these metals[…]" In the new Figure 1—figure supplement 1, however, just the opposite trend is observed. The authors should modify this statement and subsequent conclusions in the main text to reflect this fact.

4) Some of the structures shown in Figure 7 have poor resolution and are difficult to decipher.

[Editors' note: further revisions were requested prior to acceptance, as described below.]

Thank you for resubmitting your work entitled "Polymerase θ is a Robust Terminal Transferase that Oscillates Between Three Different Mechanisms During End-Joining" for further consideration at *eLife*. Your revised article has been favorably evaluated by John Kuriyan (Senior editor) and two reviewers, one of whom is a member of our Board of Reviewing Editors.

Upon initial inspection of the revision, a reviewing editor found the revisions to be largely adequate (thus the initial decision on the revised manuscript). However, one of the expert reviewers subsequently also read the revision and pointed out that the main concern remains. The reviewing editor agrees that this is the case. We apologize for this mix up, but must insist that this remaining point (and a few minor textual changes) be addressed before the paper can be accepted for publication.

Remaining point: The authors have shown in the new supplement that pol θ has template-independent polymerization activity under conditions that more closely approximate in vivo ion concentrations (2 mM Mg, 0.2 mM Mn). However, the key experiments in the paper (Figure 2 and Figure 3) are still performed at unphysiological Mn concentrations (1 mM). The authors should redo Figure 2 and Figure 3 with 50 µM Mn^2+^ and 1 mM Mg^2+^, perhaps in combination with deep sequencing of products. Even if the amount of template-independent synthesis is now less than what's seen in vivo, it would still show that under physiological salt conditions, the enzyme can exhibit this behavior.

---

## [Author Response]

*This paper provides a potentially important contribution to the elucidation of the mechanism of alt-EJ. The idea that pol theta can carry out DNA synthesis in three different modes is novel and interesting. The terminal transferase activity of pol θ may very well become an important biotechnology tool.*

We thank the reviewers for their professional and fair assessment of our findings and their insight into polymerase mechanisms. We have addressed all of the comments and concerns as listed in detail below as separate responses to each point. We have worked diligently to perform several additional experiments that address all of the reviewers comments and concerns. We are satisfied with the clear results obtained from this additional work and hope the reviewers agree that the revised manuscript is now more thorough with all the proper controls, and presents clear conclusions regarding the novel mechanisms of Polθ terminal transferase and end-joining activities.

*Essential revisions:*

*For the most part, the data are performed to a high standard. The following, major issues must be addressed.*

*1) Mutagenesis due to Mn^2+^ has been extensively documented for many polymerases of different families. This metal ion is known to interfere with polymerase fidelity as these enzymes become more proficient at making mismatches but also discriminate less against the wrong sugar moiety. Mn^2+^ is also known to activate the elongation and TdT activity of pol u. RNA polymerases are also affected: poly(A) polymerase can synthesize a poly(A) oligomer with just ATP and Mn^2+^, whereas in the presence of magnesium it necessitates an RNA primer. So it should not come as a surprise that pol θ is more efficient at incorporating ribonucleotides and modified nucleotides in the presence of Mn^2+^. The authors make a statement that the switching mechanism facilitated by Mn^2+^ is used to generate diversity during alternative end-joining* in vivo*. Going from the experiments reported in this manuscript to saying that Mn^2+^ is a cofactor* in vivo *requires a huge leap of faith. The concentrations of Mn^2+^ used in the* in vitro *experiments are well above the reported Mn^2+^ concentration in cellulo: 1 to 5 mM vs. 10-200 uM. Moreover, gel 1E shows that at 10 mM Mg^2+^ (which is in the range of Mg^2+^ concentration in the nucleus: 0.5- 10 mM) the stimulatory effect of 1 mM Mn^2+^ is essentially non existent. It is important to show that low μM amounts of Mn^2+^ support the claimed activities.*

We thank the reviewers for their insight and agree with their comments. We have discussed the cellular concentrations of manganese and magnesium with colleagues and identified references that have measured these concentrations (which are now included in the text) in cells. The references indicate that manganese and magnesium cellular concentrations are approximately 0.2 mM and 1.0 mM, respectively, which are within the range suggested by the reviewers. We have now tested whether these lower concentrations of manganese and magnesium stimulate Pol template-independent activity in Figure 1—figure supplement 1. The results clearly show that Polθ performs template-independent activity in the presence of 0.2 mM manganese and that this activity is further stimulated by both 1 mM and 2 mM concentrations of magnesium. Thus, these new data show that physiological concentrations of magnesium and manganese stimulate Polθ template-independent activity by 3-8 fold. New results described in detail below unequivocally demonstrate that this activity is indeed template-independent.

Mn^2+^ is notorious in making the DNA synthesis reaction promiscuous, and given that Pol θ has a very low fidelity (Kunkel's work), the reaction can become even more nonspecific with Mn(II). It is not ruled out that the template independent activity is not misincorporations or slippages.

We again thank the reviewers for their insight into polymerase mechanisms, and we also thought about this possibility and agree with the reviewers concern. We therefore performed several experiments to fully address this important concern. First, in new Figure 2—figure supplement 2 and 2B we performed a solid-phase experiment that allows analysis of primer-template extension following removal of all possible excess ssDNA that may be used as a template. For example, after annealing a 2-fold excess of template to the biotinylated primer to be sure all of the primer is saturated with template strand, we then immobilized the primer-template, then washed away all excess unbound ssDNA template. Next, we added Pol and performed primer extension in solid-phase in the presence of manganese, then sequenced the immobilized extended primer after removing it from the beads by boiling. If the polymerase exhibits extremely low fidelity during primer extension with manganese this would be detected in the sequence. Although we did observe some misicnorporations during primer extension, the rate was 5.6 x 10-2. The rate for frameshift mutations was lower at 6.9 x 10-3. Once the polymerase reached the end of the template, however, the sequence generated was mostly random which strongly indicates a template-independent mechanism and not the result of misincorporation or slippage during a template-dependent mechanism.

As another control for template-dependent activity in new Figure 2—figure supplement 2, we visualized the ability of Polθ to perform template-independent activity following primer extension in the presence of manganese. Here, we compared primer extension in the presence of magnesium versus manganese. The results clearly show that the polymerase is able to continue DNA synthesis far beyond the end of the template exclusively in the presence of manganese which supports the template-independent activity shown in Figure 2—figure supplement 2.

As an additional control for template-dependent activity in new Figure 2—figure supplement 2, we compared miscincoporation and mismatch extension during primer extension in the presence of a single nucleotide (dATP) versus an identical reaction with dATP, but in the absence of the template strand. For example, both reactions were performed in an identical manner with dATP, however, one reaction contained a primer-template, whereas the other reaction contained only the primer which exclusively allows for terminal transferase activity. The results clearly show that misincorporation and mismatch extension (by addition of dATP only), which are template-dependent mechanisms, occur very slowly. In contrast, the same reaction performed with the primer only resulted in rapid and long-range extension of the primer. Thus, these results demonstrate that the ssDNA extension mechanism occurs on an entirely different time scale than misincorporation and mismatch extension and is therefore is not due to misincorporation or slippage events.

As a final control for template-dependent activity in Figure 2—figure supplement 3, we tested whether Polθq exhibits de novo synthesis of DNA and RNA. For example, here we simply incubated Pol with α-32P-dCTP either in the presence of deoxy-ribonucleotides (dGTP, dATP, dTTP) or ribonucleotides (GTP, ATP, UTP) along with manganese. The results show that the polymerase exhibits de novo synthesis which confirms its ability to synthesize DNA and RNA in the absence of a template strand.

We hope the reviewers will agree that these experiments were carried out with all the proper controls and provide several lines of evidence that the polymerase exhibits template-independent activity in the presence of manganese.

*2) The authors very strongly imply that one and the same pol θ molecule (or dimer) cycles through the different modes of synthesis. Their data does not justify this conclusion. They need to repeat their in vitro end-joining analysis with sequencing under conditions where dissociation and rebinding are not allowed.*

We again thank the reviewers for their insight into polymerase mechanisms. Although the reviewers feel that the manuscript strongly implies that “one and the same pol θ molecule (or dimer) cycles through the different modes of synthesis”, we did not intend to imply this precise mechanism in the language used in the original manuscript. Nevertheless, since the polymerase appears to perform terminal transferase activity in a processive manner, we do agree that the possibility of a single polymerase (or dimers of the polymerase) switching between the three mechanisms without dissociating from the initial substrate is indeed intriguing and a formal possibility. We therefore set out to specifically test this idea as follows. First, we determined what concentration of ssDNA acts as a potent trap of the polymerase in solution in (Figure 2—figure supplement 4). We found that when Polθ was added to a combination of 50 nM of radio-labeled ssDNA and a 150-fold excess of unlabeled ssDNA, the polymerase was unable to extend the radio-labeled substrate even after 30 min, which demonstrates that a 150-fold excess of ssDNA acts as a potent trap of the polymerase when it is not pre-bound to DNA. We then used this trap to better assess the processivity of Pol terminal transferase activity in panel B of the same figure (Figure 2—figure supplement 4). Here, Polθ ssDNA extension was initiated on the radio-labeled ssDNA, then after 5 min either no trap or a 150-fold excess of cold ssDNA was added and the reaction was allowed to proceed during a time course. Remarkably, addition of the ssDNA trap had no effect Polθ terminal transferase activity, for example the time course in the presence and absence of the trap looked identical. Since the 150-fold excess ssDNA trap effectively sequesters the polymerase from solution in the control experiment in panel A, these data demonstrate that Pol terminal transferase activity is highly processive. Because the polymerase is processive during ssDNA extension and performs three different extension mechanisms during this process, these data imply that one and the same polymerase (or polymerase dimers) cycles through the different activities before dissociating from the initial ssDNA substrate.

To provide more direct evidence for this processive switching mechanism, we performed another important control experiment in new Figure 2—figure supplement 5. Here, Polθ terminal transferase activity was performed in solid-phase after excess Polθ was removed from solution. For example, after the polymerase was allowed to bind immobilized ssDNA in the absence of dNTPs, the beads were washed extensively to remove excess unbound polymerase. Next the reaction was initiated by the addition of dNTPs in buffer containing magnesium and manganese. Then after 15 seconds, 150-fold excess ssDNA or no trap was added and the reaction was allowed to proceed. After the reaction was terminated, the immobilized extended ssDNA was removed from the beads by boiling then sequenced. Consistent with the results obtained in Figure 2—figure supplement 4, the polymerase was not inhibited by the addition of the ssDNA trap. In fact, we even observed that excess ssDNA in trans appears to promote further extension of the initial ssDNA substrate by enabling the polymerase to utilize the ssDNA trap as a template in trans. We also observed that under these conditions, where the polymerase is limiting and there is an abundant amount of ssDNA in trans, that Pol readily switches between the three different terminal transferase activities. Together, these new experiments provide strong evidence that the polymerase acts processively during ssDNA extension, and that the polymerase readily switches between the three different terminal transferase activities without dissociating from the initial ssDNA template.

As a final control for this polymerase switching mechanism under conditions where the polymerase is effectively trapped if it dissociates from the initial DNA substrate, we repeated the identical alt-EJ reaction in Figure 3, but added a 150-fold excess of ssDNA 30 seconds after the reaction was initiated (new Figure 3—figure supplement 2). Here, the trap would effectively sequester the polymerase if it dissociated from the initial end-joining substrate(s) prior to completion of the reaction. We observed similar insertion products at DNA repair junctions in the presence of the trap as those obtained in Figure 3 without a trap. In fact, the median insertion tract length was slightly longer in the presence of the ssDNA trap, which is consistent with the results obtained in Figure 2—figure supplement 5. These insertion tracts, which are likely generated by a single molecule or dimer of Polθ as a result of its high processivity, also show evidence for the different mechanisms of terminal transferase activity (new Figure 3—figure supplement 2). Hence, these results indicate that the polymerase also acts processively during alt-EJ, and in doing so oscillates between the different mechanisms of terminal transferase activity which contribute to genetic diversity.

We hope the reviewers will agree that the carefully planned experiments with proper controls support the idea that Pol acts with high processivity during ssDNA extension and end-joining and therefore likely switches between the different mechanisms of terminal transferase activity prior to dissociating from the initial substrate.

[Editors' note: further revisions were requested prior to acceptance, as described below.]

*The manuscript has been greatly improved but there are some remaining issues that need to be addressed before acceptance, as outlined below:*

*1) Figure 6 seems to argue that even WT pol θ is not highly processive in the primer-template extension reaction, as addition of the DNA trap dramatically reduces observed extension. This contrasts with the reported processivity in terminal transferase activity. The authors should note and discuss this difference in the text.*

We agree with the reviewers comment and appreciate their insight. The various data presented in the paper strongly suggest that the mechanism by which Polθ extends ssDNA is very different from the mechanism by which the polymerase performs primer extension. For example, although loop 2 is not needed for primer-template extension (shown in Figure 6 and previous work: Kent, T. et al. Nat. Struct. Mol. Biol. (2015)), this motif is essential for ssDNA extension (Figure 6). Thus, the processivity of ssDNA extension compared to primer extension is also likely to be different. We therefore included a statement in the text as follows. “These data also suggest that PolθWT exhibits lower processivity during primer-template extension compared to ssDNA extension (compare Figure 6 and Figure 2—figure supplement 4).”

*Moreover, it is unclear why primer extension appears to stop at 45 nt in Figure 6 but proceeds to >100 nt in Figure 5 and Figure 6. Perhaps this is due to different reaction times, but these times are not noted in the figure itself or legend for 5D and 6D.*

Pol primer-template extension terminates at the 5’ end of the primer when Mn^2+^ is absent from the reaction and Mg^2+^ is present. This is shown clearly in Figure 2—figure supplement 2. However, when Mn^2+^ is included in the reaction, then Polθ switches to template-independent activity at the end of the template and therefore continues to extend the primer beyond the 5’ end of the template (see Figure 2—figure supplement 2). This is the case in Figure 5 and Figure 6 where Mn^2+^ is present in the reaction, but not Mg^2+^. In Figure 6, only Mg^2+^ is present resulting in primer-template termination at the end of the template. The presence of Mn^2+^ or Mg^2+^ for each experiment is clearly indicated in the figure legends.

Figure 6 is exclusively ssDNA extension with Mn^2+^ which is clearly indicated in the figure legend.

2) No statistical tests are presented.

We have now performed unpaired t tests to compare the statistical significance between the percent of insertions generated by non-templated extension versus snap-back extension mechanisms for the in vitro and in vivo data. The results of these tests indicate that the differences between the percent insertions generated by non-templated versus snap-back replication for each system are not statistically significant and this is now indicated in the figure legends for Figure 3 and Figure 4 as follows. “t test indicates no significant difference between percent of non-templated and templated in cis insertions.”

We emphasize that the quantitative analyses of the type of insertions generated by Polθ (depicted by pie charts in Figure 3 and Figure 4) were presented to show that Pol is capable of utilizing the three different modes at roughly equal proportions. For example, the text below in the manuscript describe the data presented in the pie charts:

“A median insertion length of 7 bp was observed (Figure 3), and cumulative analysis of individual nucleotide insertion events reveals a roughly equal proportion of insertions due to the three modes of terminal transferase activity identified in Figure 2, for example non-templated extension, templated extension in cis, and templated extension in trans (Figure 3).

“For example, similar to the results obtained in the in vitro alt-EJ system (Figure 3), cumulative analysis of individual nucleotide insertion events produced in vivo demonstrates that Polθ generates a roughly equal proportion of insertion events due to the three different modes of terminal transferase activity (Figure 4).”

Thus, the slight differences between the proportion of different types of insertions generated by the polymerase is not what is significant. It is the overall observation that the polymerase generates insertions by using the 3 different mechanisms of terminal transferase activity at roughly equal proportions.

*3) The authors state "We note that Mn^2+^ had a greater stimulatory effect in the absence of Mg^2+^, which shows competition between these metals[…]" In the new Figure 1—figure supplement 1, however, just the opposite trend is observed. The authors should modify this statement and subsequent conclusions in the main text to reflect this fact.*

We thank the reviewers for their close attention to detail and we agree with the results which appear to differ based on whether the divalent ion concentration is limiting or not. For example, in the new Figure 1, Mg^2+^ and Mn^2+^ together stimulate template independent activity. These are at low concentrations which may be limiting (less than or equal to 1 mM). At higher concentrations in Figure 1 (i.e. greater than or equal to 5 mM), which are probably not limiting, there seems to be competition between the ions Mg^2+^ and Mn^2+^. Thus, we removed the following general statement, "We note that Mn^2+^ had a greater stimulatory effect in the absence of Mg^2+^, which shows competition between these metals[…]".

Since Polθ DNA synthesis activity is fully supported by Mn^2+^, these data still indicate that the metal binds to the same positions as Mg^2+^. Thus, the paragraph now reads as follows: “Since Pol DNA synthesis activity is fully supported by Mn^2+^ (Figure 1, lane 25), this indicates that Mn^2+^ binds to the same positions as Mg^2+^ within the polymerase active site which is necessary for the nucleotidyl transferase reaction. Consistent with this[…]”

*4) Some of the structures shown in Figure 7 have poor resolution and are difficult to decipher.*

We thank the reviewers for pointing this out. We have now revised the figure to only include high resolution images.

[Editors' note: further revisions were requested prior to acceptance, as described below.]

Remaining point: The authors have shown in the new supplement that pol theta has template-independent polymerization activity under conditions that more closely approximate in vivo ion concentrations (2 mM Mg, 0.2 mM Mn). However, the key experiments in the paper (Figure 2 and Figure 3) are still performed at unphysiological Mn concentrations (1 mM). The authors should redo Figure 2 and Figure 3 with 50 µM Mn^2+^ and 1 mM Mg^2+^, perhaps in combination with deep sequencing of products. Even if the amount of template-independent synthesis is now less than what's seen in vivo, it would still show that under physiological salt conditions, the enzyme can exhibit this behavior.

We agree with the reviewer that it is important to test Polθ mediated end-joining and ssDNA extension using low concentrations of Mg and Mn to more closey model physiological conditions. We therefore repeated the experiments in Figure 2 and Figure 3 using 50 µM Mn^2+^ and 1 mM Mg^2+^ as requested. The results of these experiments are now included as Figure 2—figure supplement 6 and Figure 3—figure supplement 3. Consistent with results from the previous revision using low concentrations of Mg^2+^ and Mn^2+^, we observe that Polθ promotes insertions during end-joining (Figure 3—figure supplement 3) which have a median length of 9 bp (for insertions greater than 2 bp in length). These insertion lengths are similar to what we observed using higher concentrations of Mg^2+^ and Mn^2+^ in vitro (Figure 3) and in cells (Figure 4). We also show that Pol effectively extends ssDNA using these low concentrations of Mg^2+^ (1 mM) and Mn^2+^ (50 µM) and that these extension products are facilitated by both templated and non-templated DNA synthesis mechanisms, which shows that the polymerase also switches between these mechanisms during ssDNA extension under these low ion concentrations (Figure 2—figure supplement 6). These experiments taken together with similar experiments in the first manuscript revision (Figure 1—figure supplement 1) clearly show that Polθ exhibits both templated and non-templated terminal transferase activities and end-joining activity resulting in insertions under physiologically relevant concentrations of Mg^2+^ and Mn^2+^.